# NoisyNN: Exploring the Impact of Information Entropy Change in Learning Systems

## Abstract

We investigate the impact of entropy change in deep learning systems by noise injection at different levels, including the embedding space and the image. The series of models that employ our methodology are collectively known as Noisy Neural Networks (NoisyNN), with examples such as NoisyViT and NoisyCNN discussed in the paper. Noise is conventionally viewed as a harmful perturbation in various deep learning architectures, such as convolutional neural networks (CNNs) and vision transformers (ViTs), as well as different learning tasks like image classification and transfer learning. However, this work shows noise can be an effective way to change the entropy of the learning system. We demonstrate that specific noise can boost the performance of various deep models under certain conditions. We theoretically prove the enhancement gained from positive noise by reducing the task complexity defined by information entropy and experimentally show the significant performance gain in large image datasets, such as the ImageNet. Herein, we use the information entropy to define the complexity of the task. We categorize the noise into two types, positive noise (PN) and harmful noise (HN), based on whether the noise can help reduce the complexity of the task. Extensive experiments of CNNs and ViTs have shown performance improvements by proactively injecting positive noise, where we have achieved an unprecedented top 1 accuracy of 95% on ImageNet. Both theoretical analysis and empirical evidence have confirmed that the presence of positive noise can benefit the learning process, while the traditionally perceived harmful noise indeed impairs deep learning models. The different roles of noise offer new explanations for deep models on specific tasks and provide a new paradigm to improve model performance. Moreover, it reminds us that we can influence the performance of learning systems via information entropy change. Code for reproducing NoisyViT on ImageNet is available at NoisyViT.

## 1 Introduction

Noise, traditionally viewed as an obstacle in machine learning applications Thulasidasan et al. (2019), is universal due to various factors such as environmental conditions, equipment calibration, and human activities Ormiston et al. (2020). In computer vision, noise can emerge at multiple stages. During image acquisition, for instance, imaging systems may introduce noise due to factors such as detector limitations, or electronic interference. This could manifest as electronic or thermal noise, leading to random variations in pixel values or color discrepancies in the captured images Sijbers et al. (1996). In specific imaging contexts, *e.g.* fluoroscopic imaging, additional noise components may be present, including photon scattering artifacts, uncorrelated quantum noise originating from photon statistical variations, and correlated noise associated with readout processes Zhang et al. (2014). Additionally, noise can also be introduced during the image preprocessing phase. Operations such as image resizing, filtering, interpolation, or color space conversion are potential sources of noise Al-Shaykh & Mersereau (1998). For example, resizing might result in aliasing artifacts, non-linear interpolation methods, such as B-splines Hou & Andrews (1978), may lead to local morphological distortions in the image, and low-pass filtering could cause a loss of image detail and texture.

Prevailing literature typically assumes that noise adversely affects the task Sethna et al. (2001); Owotogbe et al. (2019). However, is this assumption always applicable? Our work seeks to thoroughly examine this

critical question. We recognize that the vague definition of noise contributes to the uncertainty in identifying and characterizing it. One effective way to categorize different noises is through analysis of task complexity change (Li, 2022). Leveraging the concept of task complexity, we can categorize noise into two types: positive noise (PN) and harmful noise (HN). PN reduces task complexity, whereas HN increases it, consistent with traditional views of noise.

Our work, which combines a theoretical analysis based on information theory with extensive empirical evaluation, reveals a surprising fact: **the simple injection of noise into deep neural networks, when done in a principled manner, can significantly enhance model performance**.

### 1.1 Scope and Contribution

Our work aims to investigate how various types of noise affect deep learning models. Specifically, the study focuses on three common types of noise, i.e., Gaussian noise, linear transform noise, and salt-and-pepper noise. Gaussian noise refers to random fluctuations that follow a Gaussian distribution in pixel values at the image level or latent representations. Linear transforms, on the other hand, refer to affine elementary transformations to the dataset of original images or latent representations. Salt-and-pepper noise is a kind of image distortion that adds random black or white values at the image level or to the latent representations.

This paper analyzes the impact of these types of noise on the performance of deep learning models for image classification tasks, including image classification and unsupervised domain adaptation. Two popular model families, Vision Transformers (ViTs) and Convolutional Neural Networks (CNNs), are considered in the study. Image classification, a foundational task in computer vision, involves predicting the class label of a given input image. In contrast, domain adaptation addresses scenarios where the training and test data originate from distinct distributions, commonly referred to as different domains. This comprehensive analysis aims to shed light on how noise impacts these critical tasks across different model architectures.

We start by presenting a comprehensive theoretical analysis of how these three types of noise impact deep models. Building on the theoretical foundation, we propose **NoisyNN**, a novel method designed to enhance the deep neural network performance on Image Classification and Domain Adaptation. To validate the effectiveness of NoisyNN, we conduct extensive experiments with ViTs and CNNs. Our empirical findings demonstrate the huge benefits of leveraging positive noise.

The contributions of this paper are summarized as follows:

- We re-examined the conventional view that noise, by default, has a negative impact on deep learning models. Our theoretical analysis and experimental results show that noise can be a positive support for deep learning models on image classification tasks.

- We introduce NoisyNN, an innovative approach that strategically utilizes positive noise. NoisyNN achieves state-of-the-art on various image classification and domain adaptation tasks.

- Our study, along with the success of NoisyNN, prompts a re-visit of the role of noise in machine learning. This opens new avenues for future research to enhance deep learning model performance on various learning tasks.

### 1.2 Related Work

**Positive Noise.** Common knowledge often assumes noise to be detrimental to tasks. However, empirical evidence also suggests that some forms of noise could be beneficial. For instance, within the signal processing community, it has been shown that random noise can facilitate stochastic resonance, enhancing the detection of weak signals Benzi et al. (1981). In neuroscience, noise has been recognized for its potential to boost brain functionality McClintock (2002); Mori & Kai (2002).

Recent work by Li (2022) marks a significant advance in the theoretical understanding of the characteristics of different noises. By employing information theory, they differentiate between beneficial "positive noise" and detrimental "pure noise", based on their impact on task complexity. However, their analysis has three

notable limitations: 1. it is confined to *only the image space*; 2. all experiments are conducted *only on shallow models*, far from the current best practices 3. it does not answer the practical question: *how to create and leverage positive noise?*

Our study aims to address these limitations. We make a significant extension to the positive-noise framework proposed by Li (2022). Our work not only confirms the presence of positive and harmful noise in embedding space but more importantly demonstrates that *leveraging positive noise in deeper layers of the embedding space is often more effective* (see Fig 2 b&d). Furthermore, we propose a practical approach to leverage the positive noise in deep models, we term it "NoisyNN". This approach promises to unlock new potentials in the application of noise to enhance neural network performance.

**Data Augmentation.** Data augmentation has been playing an important role in training deep vision models, as evidenced by its demonstrated effectiveness (Yang et al., 2023b). The general idea of data augmentation is to compose transformation operations that can be applied to the original data $x$ to create transformed data $x'$ without severely altering the semantics. Common data augmentation range from simple techniques like random flip and crop (Krizhevsky et al., 2012) to more complex techniques like MixUp (Zhang et al., 2017), CutOut (DeVries & Taylor, 2017), AutoAugment (Cubuk et al., 2019), AugMix (Hendrycks et al., 2019) RandAugment (Cubuk et al., 2020) among others. More comprehensive reviews can be found in (Mumuni & Mumuni, 2022). The proposed approach is related to the research on data augmentation but stands apart due to its theoretical foundation. Our framework provides a more controlled and principled way to augment data, setting it apart from conventional methods, which often require substantial domain knowledge and ad-hoc design, as noted in (Cubuk et al., 2020).

**Comparison with Manifold MixUp.** Our NoisyNN shares some similarities with Manifold MixUp (Verma et al., 2019), a regularization technique designed for supervised image classification that extends the MixUp strategy to the embedding space by linearly interpolating embedding vectors $z_i$ (instead of images $x_i$) along with their corresponding labels $y_i$. However, there are several key differences. Unlike Manifold MixUp, which aims to flatten class representations through training on interpolated synthetic samples, our NoisyNN is grounded in a theoretical analysis of how noise injection impacts task complexity in view of information entropy, as introduced by (Li, 2022).

Additionally, we derived the optimal form of noise injection (Eq.20) within the linear transform noise design space, which Manifold MixUp does not provide. Procedurally, Manifold MixUp interpolates both embeddings and labels to generate synthetic samples, followed by training on these samples, as its theoretical foundation relies on modifying both features and labels. In contrast, our method perturbs only the embeddings and leaves the labels unchanged, as our theoretical analysis is based on un-interpolated labels. Investigating whether label interpolation could be integrated into our theoretical framework may be a promising avenue for future research.

## 2 Preliminary

In information theory, the entropy Shannon (2001) of a random variable $x$ is defined as:

$$H(x) = \begin{cases} -\int p(x) \log p(x) dx & \text{if } x \text{ is continuous} \\ -\sum_x p(x) \log p(x) & \text{if } x \text{ is discrete} \end{cases} \tag{1}$$

where $p(x)$ is the distribution of the given variable $x$. The mutual information of two random discrete variables $(x, y)$ is denoted as Cover (1999):

$$\begin{aligned} MI(x, y) =& D_{KL}(p(x, y) \parallel p(x) \otimes p(y)) \\ =& H(x) - H(x|y) \end{aligned} \tag{2}$$

where $D_{KL}$ is the Kullback-Leibler divergence Kullback & Leibler (1951), and $p(x, y)$ is the joint distribution. The conditional entropy is defined as:

$$H(x|y) = -\sum p(x, y) \log p(x|y) \tag{3}$$

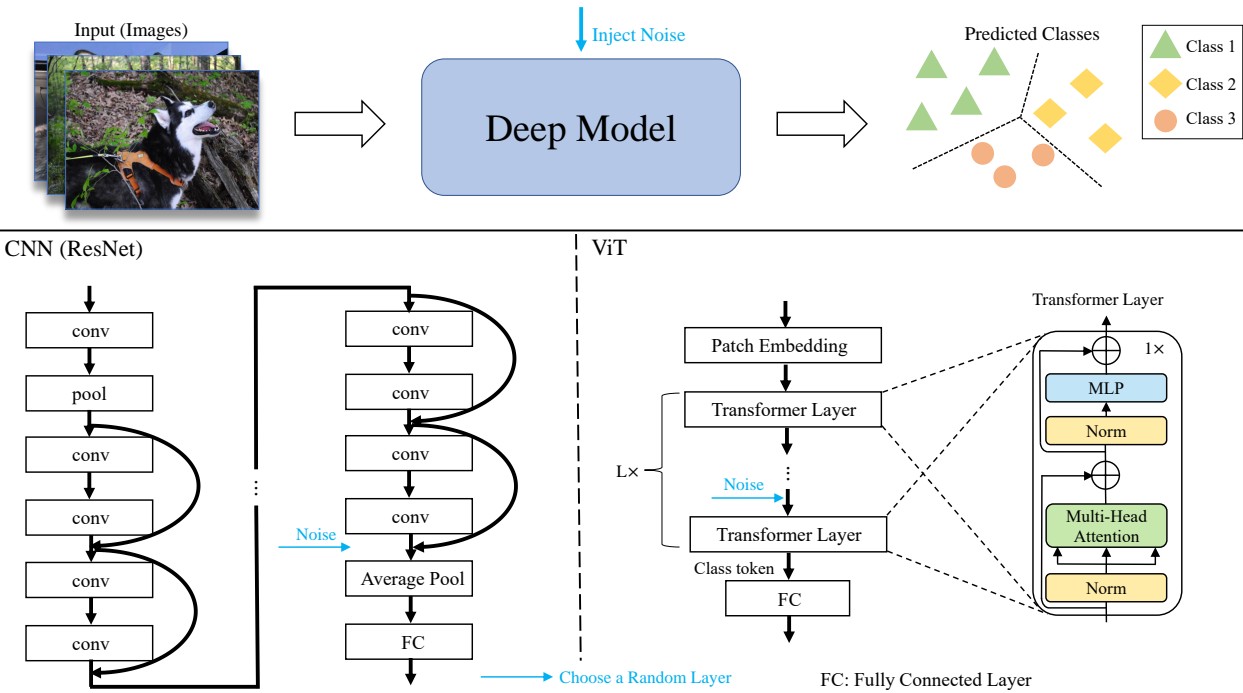

Figure 1: An overview of the proposed method: Above the black line is the standard pipeline for image classification, where the deep model can be either a CNN or a ViT. Noise is injected into a randomly selected layer of the model, as indicated by the blue arrow. Once the layer is selected, it remains fixed during both training and inference.

The above definitions can be readily expanded to encompass continuous variables through the substitution of the sum operator with the integral symbol. In this work, the noise is denoted by $\boldsymbol{\epsilon}$ if without any specific statement.

Before delving into the correlation between task and noise, it is imperative to address the initial crucial query of the mathematical measurement of a task $\mathcal{T}$. With the assistance of information theory, the complexity associated with a given task $\mathcal{T}$ can be measured in terms of the entropy of $\mathcal{T}$. Therefore, we can borrow the concepts of information entropy to explain the difficulty of the task. For example, a smaller $H(\mathcal{T})$ means an easier task and vice versa.

Since the entropy of task $\mathcal{T}$ is formulated, it is not difficult to define the entropy change when additional noise $\boldsymbol{\epsilon}$ is present Li (2022),

$$\triangle S(\mathcal{T}, \boldsymbol{\epsilon}) = H(\mathcal{T}) - H(\mathcal{T}|\boldsymbol{\epsilon}) \tag{4}$$

Formally, if the noise can help reduce the complexity of the task, i.e., $H(\mathcal{T}|\boldsymbol{\epsilon}) < H(\mathcal{T})$ then the noise has positive support. Therefore, a noise $\boldsymbol{\epsilon}$ is defined as **positive noise** (PN) when the noise satisfies $\triangle S(\mathcal{T}, \boldsymbol{\epsilon}) > 0$. On the contrary, when $\triangle S(\mathcal{T}, \boldsymbol{\epsilon}) \leq 0$, the noise is considered as the conventional noise and named **harmful noise** (HN).

$$\begin{cases} \triangle S(\mathcal{T}, \boldsymbol{\epsilon}) > 0 & \boldsymbol{\epsilon} \text{ is positive noise} \\ \triangle S(\mathcal{T}, \boldsymbol{\epsilon}) \leq 0 & \boldsymbol{\epsilon} \text{ is harmful noise} \end{cases} \tag{5}$$

**Moderate Model Assumption**: Positive noise may not be effective for deep models facing severe issues, such as overfitting, where the model starts to memorize random fluctuations in the data rather than learning the underlying patterns. In such cases, positive noise is unlikely to provide meaningful support in improving the model's performance.

## 3 Methods

The idea of exploring the influence of noise on deep models is straightforward: we add noise to the latent representations. The framework is depicted in Fig. 1. This is a universal framework for classification tasks where there are different options for deep models, such as CNNs and ViTs. Through the simple operation of injecting noise into a randomly selected layer, a model has the potential to gain additional information to reduce task complexity, thereby improving its performance. It is sufficient to inject noise into a single layer instead of multiple layers since it imposes a regularization on other layers of the model simultaneously.

For a classification problem, the dataset $(\boldsymbol{X}, \boldsymbol{Y})$ can be regarded as samplings derived from $D_{\mathcal{X}, \mathcal{Y}}$, where $D_{\mathcal{X}, \mathcal{Y}}$ is some unknown joint distribution of data points and labels from feasible space $\mathcal{X}$ and $\mathcal{Y}$, i.e., $(\boldsymbol{X}, \boldsymbol{Y}) \sim D_{\mathcal{X}, \mathcal{Y}}$ Shalev-Shwartz & Ben-David (2014). Hence, given a set of $k$ data points $\boldsymbol{X} = \{X_1, X_2, ..., X_k\}$, the label set $\boldsymbol{Y} = \{Y_1, Y_2, ..., Y_k\}$ is regarded as sampling from $\boldsymbol{Y} \sim D_{\mathcal{Y}|\mathcal{X}}$. The complexity of $\mathcal{T}$ on dataset $\boldsymbol{X}$ is formulated as:

$$H(\mathcal{T}; \boldsymbol{X}) = H(\boldsymbol{Y}; \boldsymbol{X}) - H(\boldsymbol{X}) \tag{6}$$

Accordingly, the operation of adding noise at the image level can be formulated as Li (2022):

$$\begin{cases} H(\mathcal{T}; \boldsymbol{X} + \boldsymbol{\epsilon}) = -\sum_{\boldsymbol{Y} \in \mathcal{Y}} p(\boldsymbol{Y}|\boldsymbol{X} + \boldsymbol{\epsilon}) \log p(\boldsymbol{Y}|\boldsymbol{X} + \boldsymbol{\epsilon}) & \boldsymbol{\epsilon} \text{ is additive noise} \\ H(\mathcal{T}; \boldsymbol{X}\boldsymbol{\epsilon}) = -\sum_{\boldsymbol{Y} \in \mathcal{Y}} p(\boldsymbol{Y}|\boldsymbol{X}\boldsymbol{\epsilon}) \log p(\boldsymbol{Y}|\boldsymbol{X}\boldsymbol{\epsilon}) & \boldsymbol{\epsilon} \text{ is multiplicative noise} \end{cases} \tag{7}$$

Inspired by the previous work Li (2022), given a set of $k$ image embeddings $\boldsymbol{Z} = \{Z_1, Z_2, ..., Z_k\}$, the label set $\boldsymbol{Y} = \{Y_1, Y_2, ..., Y_k\}$ is regarded as sampling from $\boldsymbol{Y} \sim D_{\mathcal{Y}|\mathcal{Z}}$. The complexity of $\mathcal{T}$ on dataset $\boldsymbol{Z}$ is formulated as:

$$H(\mathcal{T}; \boldsymbol{Z}) \coloneqq H(\boldsymbol{Y}; \boldsymbol{Z}) - H(\boldsymbol{Z}) \tag{8}$$

The operation of proactively injecting noise in the latent space is defined as:

$$\begin{cases} H(\mathcal{T}; \boldsymbol{Z} + \boldsymbol{\epsilon}) \coloneqq H(\boldsymbol{Y}; \boldsymbol{Z} + \boldsymbol{\epsilon}) - H(\boldsymbol{Z}) & \boldsymbol{\epsilon} \text{ is additive noise} \\ H(\mathcal{T}; \boldsymbol{Z}\boldsymbol{\epsilon}) \coloneqq H(\boldsymbol{Y}; \boldsymbol{Z}\boldsymbol{\epsilon}) - H(\boldsymbol{Z}) & \boldsymbol{\epsilon} \text{ is multiplicative noise} \end{cases} \tag{9}$$

where $\boldsymbol{Z}$ are the embeddings of the images. The definition of Eq. 9 differs from the conventional definition, as our method injects the noise into the latent representations instead of the original images. The Gaussian noise is additive, the linear transform noise is also additive, while the salt-and-pepper is multiplicative.

**Gaussian Noise** The Gaussian noise is one of the most common additive noises that appear in computer vision tasks. The Gaussian noise is independent and stochastic, obeying the Gaussian distribution. Without loss of generality, defined as $\mathcal{N}(\mu, \sigma^2)$. Since our injection happens in the latent space, therefore, the complexity of the task is:

$$H(\mathcal{T}; \boldsymbol{Z} + \boldsymbol{\epsilon}) = H(\boldsymbol{Y}; \boldsymbol{Z} + \boldsymbol{\epsilon}) - H(\boldsymbol{Z}). \tag{10}$$

We assume that both $\boldsymbol{Z}$ and $\boldsymbol{Y}$ follow a multivariate normal distribution. Additionally, we can transform the distributions of $\boldsymbol{Z}$ and $\boldsymbol{Y}$ to make them (approximately) follow the multivariate normal distribution, even if they initially do not Box & Cox (1964) Feng et al. (2014). According to the definition in Equation 4, the entropy change with Gaussian noise is:

$$\begin{aligned} \triangle S(\mathcal{T}, \boldsymbol{\epsilon}) =& H(\boldsymbol{Y}; \boldsymbol{Z}) - H(\boldsymbol{Z}) - (H(\boldsymbol{Y}; \boldsymbol{Z} + \boldsymbol{\epsilon}) - H(\boldsymbol{Z})) \\ =& H(\boldsymbol{Y}; \boldsymbol{Z}) - H(\boldsymbol{Y}; \boldsymbol{Z} + \boldsymbol{\epsilon}) \\ =& \frac{1}{2} \log \frac{|\boldsymbol{\Sigma_Z}||\boldsymbol{\Sigma_Y} - \boldsymbol{\Sigma_{YZ}}\boldsymbol{\Sigma_Z^{-1}}\boldsymbol{\Sigma_{ZY}}|}{|\boldsymbol{\Sigma_{Z+\epsilon}}||\boldsymbol{\Sigma_Y} - \boldsymbol{\Sigma_{YZ}}\boldsymbol{\Sigma_{Z+\epsilon}^{-1}}\boldsymbol{\Sigma_{ZY}}|} \\ =& \frac{1}{2} \log \frac{1}{(1 + \sigma_\epsilon^2 \sum_{i=1}^k \frac{1}{\sigma_{Z_i}^2})(1 + \lambda \sum_{i=1}^k \frac{\text{cov}^2(Z_i, Y_i)}{\sigma_{X_i}^2(\sigma_{Z_i}^2 \sigma_{Y_i}^2 - \text{cov}^2(Z_i, Y_i))})} \end{aligned} \tag{11}$$

where $\lambda = \frac{\sigma_\epsilon^2}{1 + \sum_{i=1}^k \frac{1}{\sigma_{Z_i}^2}}$, $\sigma_\epsilon^2$ is the variance of the Gaussian noise, $\text{cov}(Z_i, Y_i)$ is the covariance of sample pair $X_i, Y_i$, $\sigma_{Z_i}^2$ and $\sigma_{Y_i}^2$ are the variance of data sample $Z_i$ and data label $Y_i$, respectively. The detailed

derivations can be found in section 1.1.2 of the supplementary. Given a dataset, the variance of the Gaussian noise, and statistical properties of data samples and labels control the entropy change, we define the function:

$$
\begin{aligned}
M =& 1 - (1 + \sigma_\epsilon^2 \textstyle\sum_{i=1}^k \frac{1}{\sigma_{Z_i}^2})(1 + \lambda \sum_{i=1}^k \frac{\text{cov}^2(Z_i, Y_i)}{\sigma_{Z_i}^2(\sigma_{Z_i}^2 \sigma_{Y_i}^2 - \text{cov}^2(Z_i, Y_i))}) \\
=& -\sigma_\epsilon^2 \textstyle\sum_{i=1}^k \frac{1}{\sigma_{Z_i}^2} - \sigma_\epsilon^2 \sum_{i=1}^k \frac{1}{\sigma_{Z_i}^2} \cdot \lambda \sum_{i=1}^k \frac{\text{cov}^2(Z_i, Y_i)}{\sigma_{Z_i}^2(\sigma_{Z_i}^2 \sigma_{Y_i}^2 - \text{cov}^2(Z_i, Y_i))} - \lambda \sum_{i=1}^k \frac{\text{cov}^2(Z_i, Y_i)}{\sigma_{Z_i}^2(\sigma_{Z_i}^2 \sigma_{Y_i}^2 - \text{cov}^2(Z_i, Y_i))}
\end{aligned}
\tag{12}
$$

Since $\epsilon^2 \geq 0$ and $\lambda \geq 0$, $\sigma_{Z_i}^2 \sigma_{Y_i}^2 - \text{cov}^2(Z_i, Y_i) = \sigma_{Z_i}^2 \sigma_{Y_i}^2(1 - \rho_{Z_i Y_i}^2) \geq 0$, where $\rho_{Z_i Y_i}$ is the correlation coefficient, the sign of $M$ is negative. Consequently, we conclude that **the injection of Gaussian noise into the embedding space is harmful to the task**. Detailed derivations can be found in App sec. B.

**Linear Transform Noise.** This type of noise is obtained by elementary transformation of the features matrix, i.e., $\boldsymbol{\epsilon} = Q\boldsymbol{X}$, where $Q$ is a linear transformation matrix. We name the $Q$ the quality matrix since it controls the property of linear transform noise and determines whether positive or harmful. In the linear transform noise injection in the latent space case, the complexity of the task is:

$$
H(\mathcal{T}; \boldsymbol{Z} + Q\boldsymbol{Z}) = H(\boldsymbol{Y}; \boldsymbol{Z} + Q\boldsymbol{Z}) - H(\boldsymbol{Z})
\tag{13}
$$

The entropy change is then formulated as:

$$
\begin{aligned}
\triangle S(\mathcal{T}, Q\boldsymbol{Z}) =& H(\boldsymbol{Y}; \boldsymbol{Z}) - H(\boldsymbol{Z}) - (H(\boldsymbol{Y}; \boldsymbol{Z} + Q\boldsymbol{Z}) - H(\boldsymbol{Z})) \\
=& H(\boldsymbol{Y}; \boldsymbol{Z}) - H(\boldsymbol{Y}; \boldsymbol{Z} + Q\boldsymbol{Z}) \\
=& \frac{1}{2} \log \frac{|\boldsymbol{\Sigma_Z}||\boldsymbol{\Sigma_Y} - \boldsymbol{\Sigma_{YZ}}\boldsymbol{\Sigma_Z^{-1}}\boldsymbol{\Sigma_{ZY}}|}{|\boldsymbol{\Sigma_{(I+Q)Z}}||\boldsymbol{\Sigma_Y} - \boldsymbol{\Sigma_{YZ}}\boldsymbol{\Sigma_Z^{-1}}\boldsymbol{\Sigma_{XY}}|} \\
=& \frac{1}{2} \log \frac{1}{|I+Q|^2} \\
=& -\log|I+Q|
\end{aligned}
\tag{14}
$$

Since we want the entropy change to be greater than 0, we can formulate Equation 14 as an optimization problem:

$$
\begin{aligned}
\max_Q & \triangle S(\mathcal{T}, Q\boldsymbol{Z}) \\
s.t. \ & rank(I + Q) = k \\
& Q \sim I \\
& [I+Q]_{ii} \geq [I+Q]_{ij}, i \neq j \\
& \|[I+Q]_i\|_1 = 1
\end{aligned}
\tag{15}
$$

where $\sim$ means the row equivalence. The key to determining whether the linear transform is positive noise or not lies in the matrix of $Q$. The most important step is to ensure that $I + Q$ is reversible, which is $|(I + Q)| \neq 0$. The third constraint is to make the trained classifier get enough information about a specific image and correctly predict the corresponding label. For example, for an image $X_1$ perturbed by another image $X_2$, the classifier obtained dominant information from $X_1$ so that it can predict the label $Y_1$. However, if the perturbed image $X_2$ is dominant, the classifier can hardly predict the correct label $Y_1$ and is more likely to predict as $Y_2$. The fourth constraint is to maintain the norm of latent representations. More in-depth discussion and linear transform noise added to the image level are provided in the supplementary.

**Salt-and-pepper Noise.** The salt-and-pepper noise is a common multiplicative noise for images. The image can exhibit unnatural changes, such as black pixels in bright areas or white pixels in dark areas, specifically as a result of the signal disruption caused by sudden strong interference or bit transmission

errors. In the Salt-and-pepper noise case, the entropy change is:

$$
\begin{aligned}
\triangle S(\mathcal{T}, \boldsymbol{\epsilon}) =& H(\boldsymbol{Y}; \boldsymbol{Z}) - H(\boldsymbol{Z}) - (H(\boldsymbol{Y}; \boldsymbol{Z}\boldsymbol{\epsilon}) - H(\boldsymbol{Z})) \\
=& H(\boldsymbol{Y}; \boldsymbol{Z}) - H(\boldsymbol{Y}; \boldsymbol{Z}\boldsymbol{\epsilon}) \\
=& -\sum_{\boldsymbol{Z} \in \mathcal{Z}} \sum_{\boldsymbol{Y} \in \mathcal{Y}} p(\boldsymbol{Z}, \boldsymbol{Y}) \log p(\boldsymbol{Z}, \boldsymbol{Y}) + \sum_{\boldsymbol{Z} \in \mathcal{Z}} \sum_{\boldsymbol{Y} \in \mathcal{Y}} \sum_{\boldsymbol{\epsilon} \in \mathcal{E}} p(\boldsymbol{Z}\boldsymbol{\epsilon}, \boldsymbol{Y}) \log p(\boldsymbol{Z}\boldsymbol{\epsilon}, \boldsymbol{Y}) \\
=& \mathbb{E} \left[ \log \frac{1}{p(\boldsymbol{Z}, \boldsymbol{Y})} \right] - \mathbb{E} \left[ \log \frac{1}{p(\boldsymbol{Z}\boldsymbol{\epsilon}, \boldsymbol{Y})} \right] \\
=& \mathbb{E} \left[ \log \frac{1}{p(\boldsymbol{Z}, \boldsymbol{Y})} \right] - \mathbb{E} \left[ \log \frac{1}{p(\boldsymbol{Z}, \boldsymbol{Y})} \right] - \mathbb{E} \left[ \log \frac{1}{p(\boldsymbol{\epsilon})} \right] \\
=& - H(\boldsymbol{\epsilon})
\end{aligned}
\tag{16}
$$

Obviously, the entropy change is smaller than 0, which indicates the complexity is increasing when injecting salt-and-pepper noise into the deep model. As can be foreseen, the salt-and-pepper noise is pure detrimental noise. More details and Salt-and-pepper added to the image level are in App. D.

## 4  Experiments

In this section, we conduct extensive experiments to explore the influence of various types of noises on deep learning models. We employ popular deep learning architectures, including both CNNs and ViTs, and show that the two kinds of deep models can benefit from the positive noise. We employ deep learning models of various scales, including ViT-Tiny (ViT-T), ViT-Small (ViT-S), ViT-Base (ViT-B), and ViT-Large (ViT-L) for Vision Transformers (ViTs), and ResNet-18, ResNet-34, ResNet-50, and ResNet-101 for ResNet architecture. The details of deep models are presented in the supplementary. Without specific instructions, the noise is injected at the last layer of the deep models. Note that for ResNet models, the number of macro layers is 4, and for each macro layer, different scale ResNet models have different micro sublayers. For example, for ResNet-18, the number of macro layers is 4, and for each macro layer, the number of micro sublayers is 2. The noise is injected at the last micro sublayer of the last macro layer for ResNet models. More experimental settings for ResNet and ViT are detailed in the supplementary. **While this work primarily focuses on image classification and domain adaptation, we additionally explored other related tasks: Domain Generalization (App F.8) and Text Classification (App F.9) to assess broader applicability of NoisyNN.**

### 4.1  Noise Setting

We utilize the standard normal distribution to generate Gaussian noise in our experiments, ensuring that the noise has zero mean and unit variance. Gaussian noise can be expressed as:

$$
\epsilon \sim \mathcal{N}(0, 1)
\tag{17}
$$

For linear transform noise, many possible quality $Q$ matrices could satisfy these constraints, forming a design space. Here, we adopt a simple concrete construction of $Q$ that we call a *circular shift* as a working example. In this construction, each original $Z_i$ is perturbed slightly by its immediate next neighbor $Z_{i+1}$. We can formally express the circular shift noise injection strategy as follows: Let the scalar hyperparameter $\alpha \in [0, 1]$ define the perturbation strength. The quality matrix $Q$ is implemented as $Q = \alpha * U - \alpha * I$, where $U_{i,j} = \delta_{i+1,j}$ with $\delta_{i+1,j}$ representing the Kronecker delta indicator Frankel (2011), and employing wrap-around (or "circular") indexing. The concrete formation of the quality matrix in circular shift form is then formulated as:

$$
Q = \begin{bmatrix}
-\alpha & \alpha & 0 & 0 & 0 \\
0 & -\alpha & \alpha & 0 & 0 \\
0 & 0 & -\alpha & \ddots & 0 \\
0 & 0 & 0 & \ddots & \alpha \\
\alpha & 0 & 0 & 0 & -\alpha
\end{bmatrix}
\tag{18}
$$

Table 1: ResNet with different kinds of noise on ImageNet. Vanilla means the vanilla model without noise injection. Accuracy is shown in percentage. Gaussian noise used here is subjected to standard normal distribution. Linear transform noise used in this table is designed to be positive noise. The difference is shown in the bracket.

| Model | ResNet-18 | ResNet-34 | ResNet-50 | ResNet-101 |
|---|---|---|---|---|
| Vanilla | 63.90 (+0.00) | 66.80 (+0.00) | 70.00 (+0.00) | 70.66 (+0.00) |
| + Gaussian Noise | 62.35 (-1.55) | 65.40 (-1.40) | 69.62 (-0.33) | 70.10 (-0.56) |
| + Linear Transform Noise | **79.62 (+15.72)** | **80.05 (+13.25)** | **81.32 (+11.32)** | **81.91 (+11.25)** |
| + Salt-and-pepper Noise | 55.45 (-8.45) | 63.36 (-3.44) | 45.89 (-24.11) | 52.96 (-17.70) |

Table 2: ViT with different kinds of noise on ImageNet. Vanilla means the vanilla model without injecting noise. Accuracy is shown in percentage. Gaussian noise used here is subjected to standard normal distribution. Linear transform noise used in this table is designed to be positive noise. The difference is shown in the bracket.

| Model | ViT-T | ViT-S | ViT-B | ViT-L |
|---|---|---|---|---|
| Vanilla | 79.34 (+0.00) | 81.88 (+0.00) | 84.33 (+0.00) | 88.64 (+0.00) |
| + Gaussian Noise | 79.10 (-0.24) | 81.80 (-0.08) | 83.41 (-0.92) | 85.92 (-2.72) |
| + Linear Transform Noise | **80.69 (+1.35)** | **87.27 (+5.39)** | **89.99 (+5.66)** | **88.72 (+0.08)** |
| + Salt-and-pepper Noise | 78.64 (-0.70) | 81.75 (-0.13) | 82.40 (-1.93) | 85.15 (-3.49) |

where the parameter $\alpha$ represents the linear transform strength.

For salt-and-pepper noise, we also use the parameter $\alpha$ to control the probability of the emergence of salt-and-pepper noise, which can be formulated as:

$$\begin{cases} max(Z) & \text{if } p < \alpha/2 \\ min(Z) & \text{if } p > 1 - \alpha/2 \end{cases} \quad (19)$$

where $p$ is a probability generated by a random seed, $\alpha \in [0, 1)$, and $Z$ is the latent representation of an image.

## 4.2 Image Classification Results

We implement extensive experiments on large-scale datasets such as ImageNet Deng et al. (2009) and small-scale datasets such as TinyImageNet Le & Yang (2015a) using ResNets and ViTs.

Table 3: Comparison between Positive Noise Empowered ViT with other ViT variants. Top 1 Accuracy is shown in percentage. Here, PN refers to positive noise, specifically a linear transformation designed to be positive noise.

| Model | Top1 Acc. | Params. | Image Res. | Pretrained Dataset |
|---|---|---|---|---|
| ViT-B Dosovitskiy et al. (2020) | 84.33 | 86M | $224 \times 224$ | ImageNet 21k |
| DeiT-B Touvron et al. (2021) | 85.70 | 86M | $224 \times 224$ | ImageNet 21k |
| SwinTransformer-B Liu et al. (2021) | 86.40 | 88M | $384 \times 384$ | ImageNet 21k |
| DaViT-B Ding et al. (2022) | 86.90 | 88M | $384 \times 384$ | ImageNet 21k |
| MaxViT-B Tu et al. (2022) | 88.82 | 119M | $512 \times 512$ | JFT-300M (Private) |
| ViT-22B Dehghani et al. (2023) | 89.51 | 21743M | $224 \times 224$ | JFT-4B (Private) |
| NoisyViT-B (ViT-B+PN) | **89.99** | 86M | $224 \times 224$ | ImageNet 21k |
| NoisyViT-B (ViT-B+PN) | **91.37** | 348M | $384 \times 384$ | ImageNet 21k |

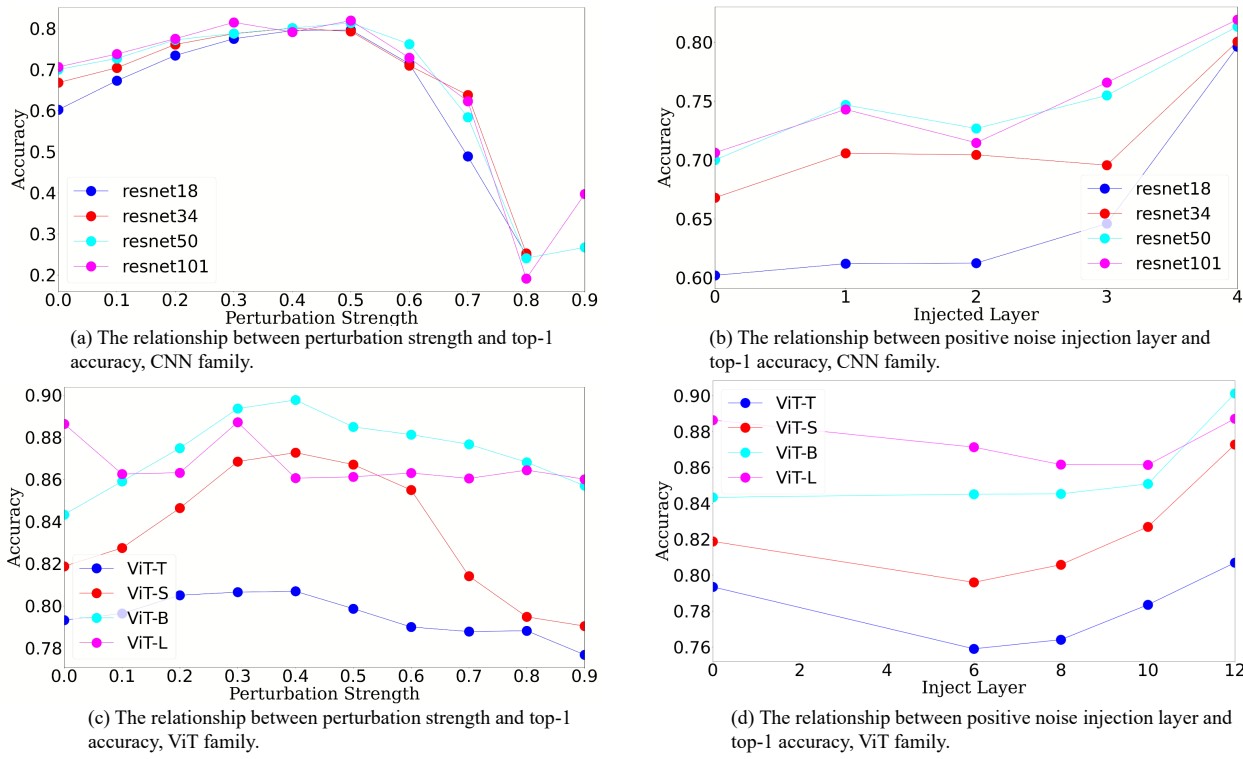

Figure 2: The relationship between the linear transform noise strength and the top 1 accuracy, and between the injected layer and top 1 accuracy. Parts (a) and (b) are the results of the CNN family, while parts (c) and (d) are the results of the ViT family. For parts (a) and (c) the linear transform noise is injected at the last layer. For parts (b) and (d), the influence of positive noise on different layers is shown. Layers 6, 8, 10, and 12 in the ViT family are chosen for the ablation study.

### 4.2.1 CNN Family

The results of ResNets with different noises on ImageNet are in Table 1. As shown in the table, with the design of linear transform noise to be positive noise (PN), ResNet improves the classification accuracy by a large margin. While the salt-and-pepper, which is theoretically harmful noise (HN), degrades the models. Note we did not utilize data augmentation techniques for ResNet experiments. The significant results show that positive noise can effectively improve classification accuracy by reducing task complexity.

### 4.2.2 ViT Family

The results of ViT with different noises on ImageNet are in Table 2. Since the ViT-L is overfitting on the ImageNet Dosovitskiy et al. (2020) Steiner et al. (2021), the positive noise did not work well on the ViT-L. As shown in the table, the existence of positive noise improves the classification accuracy of ViT by a large margin compared to vanilla ViT. The comparisons with previously published works, such as DeiT Touvron et al. (2021), SwinTransformer Liu et al. (2021), DaViT Ding et al. (2022), and MaxViT Tu et al. (2022), are shown in Table 3, and our positive noise-empowered ViT achieved the new state-of-the-art result. Note that the JFT-300M and JFT-4B datasets are private and not publicly available Sun et al. (2017), and we believe that ViT large and above will benefit from positive noise significantly if trained on larger datasets, which is theoretically supported in section 4.4.

Table 4: Top 1 accuracy on ImageNet V2 with positive linear transform noise that is designed to be positive noise.

| Model | Top1 Acc. | Params. | Image Res. |
|---|---|---|---|
| ViT-B | **73.9** | 86M | $224 \times 224$ |
| NoisyViT-B | **82.2** | 86M | $224 \times 224$ |
| NoisyViT-B | **84.8** | 348M | $384 \times 384$ |

### 4.3 Ablation Study

We also proactively inject noise into variants of ViT, such as DeiT Touvron et al. (2021), Swin Transformer Liu et al. (2021), BEiT Bao et al. (2021), and ConViT d'Ascoli et al. (2021), and the results show that positive noise could benefit various variants of ViT by improving classification accuracy significantly. The results of injecting noise to variants of ViT are reported in the supplementary Table 10. We also conducted ablation studies on the strength of linear transform noise and the injected layer. The results are shown in Fig. 2. We can observe that the deeper layer the positive noise injects, the better prediction performance the model can obtain. There are reasons behind this phenomenon. First, the latent features of input in the deeper layer have more abstract representations than those in shallow layers; second, injection to shallow layers reduces less task complexity because of trendy replacing Equation 8 with Equation 7. More results on the small dataset TinyImageNet can be found in the supplementary.

Additionally, we tested the positive linear transformation noise on another popular dataset, the ImageNet V2 Recht et al. (2019). The corresponding results are reported in Table 4.

### 4.4 Optimal Quality Matrix

As shown in Equation 15, it is interesting to learn about the optimal quality matrix of $Q$ that maximizes the entropy change while satisfying the constraints. This equals minimizing the determinant of the matrix sum of $I$ and $Q$. Here, we directly give out the optimal quality matrix of $Q$ as:

$$Q_{optimal} = \mathrm{diag}\left(\frac{1}{k+1} - 1, \ldots, \frac{1}{k+1} - 1\right) + \frac{1}{k+1}\mathbf{1}_{k \times k} \tag{20}$$

where $k$ is the number of data samples. And the corresponding upper boundary of the entropy change as:

$$\triangle S(\mathcal{T}, Q_{optimal}\boldsymbol{X}) = (k-1)\log(k+1) \tag{21}$$

The details are provided in the supplementary. We find that the upper boundary of the entropy change of injecting positive noise is determined by the number of data samples, i.e., the scale of the dataset. Therefore, the larger the dataset, the better effect of injecting positive noise into deep models. With the optimal quality matrix and the top 1 accuracy of ViT-B on ImageNet can be further improved to 95%, which is shown in Table 5.

Table 5: Top 1 accuracy on ImageNet with the optimal quality matrix of linear transform noise.

| Model | Top1 Acc. | Params. | Image Res. | Pretrained Dataset |
|---|---|---|---|---|
| NoisyViT-B+Optimal Q | **93.87** | 86M | $224 \times 224$ | ImageNet 21k |
| NoisyViT-B+Optimal Q | **95.65** | 348M | $384 \times 384$ | ImageNet 21k |

### 4.5 Domain Adaption Results

Unsupervised domain adaptation (UDA) aims to learn transferable knowledge across the source and target domains with different distributions Pan & Yang (2009) Wei et al. (2018). Recently, transformer-based methods achieved SOTA results on UDA, therefore, we evaluate the ViT-B with the positive noise on widely

Table 6: Comparison with various ViT-based methods, *i.e.* ViT-B Dosovitskiy et al. (2020), TVT-B Yang et al. (2023a), CDTrans-B Xu et al. (2022) and SSRT-B Sun et al. (2022) on **Office-Home**.

| Method | Ar2Cl | Ar2Pr | Ar2Re | Cl2Ar | Cl2Pr | Cl2Re | Pr2Ar | Pr2Cl | Pr2Re | Re2Ar | Re2Cl | Re2Pr | Avg. |
|---|---|---|---|---|---|---|---|---|---|---|---|---|---|
| ViT-B | 54.7 | 83.0 | 87.2 | 77.3 | 83.4 | 85.6 | 74.4 | 50.9 | 87.2 | 79.6 | 54.8 | 88.8 | 75.5 |
| TVT-B | 74.9 | 86.8 | 89.5 | 82.8 | 88.0 | 88.3 | 79.8 | 71.9 | 90.1 | 85.5 | 74.6 | 90.6 | 83.6 |
| CDTrans-B | 68.8 | 85.0 | 86.9 | 81.5 | 87.1 | 87.3 | 79.6 | 63.3 | 88.2 | 82.0 | 66.0 | 90.6 | 80.5 |
| SSRT-B | 75.2 | 89.0 | 91.1 | 85.1 | 88.3 | 90.0 | 85.0 | 74.2 | 91.3 | 85.7 | 78.6 | 91.8 | 85.4 |
| NoisyTVT-B | **78.3** | **90.6** | **91.9** | **87.8** | **92.1** | **91.9** | **85.8** | **78.7** | **93.0** | **88.6** | **80.6** | **93.5** | **87.7** |

Table 7: Comparison with various ViT-based methods on **Visda2017**.

| Method | plane | bcycl | bus | car | horse | knife | mcycl | person | plant | sktbrd | train | truck | Avg. |
|---|---|---|---|---|---|---|---|---|---|---|---|---|---|
| ViT-B Dosovitskiy et al. (2020) | 97.7 | 48.1 | 86.6 | 61.6 | 78.1 | 63.4 | 94.7 | 10.3 | 87.7 | 47.7 | 94.4 | 35.5 | 67.1 |
| TVT-B Yang et al. (2023a) | 92.9 | 85.6 | 77.5 | 60.5 | 93.6 | 98.2 | 89.4 | 76.4 | 93.6 | 92.0 | 91.7 | 55.7 | 83.9 |
| CDTrans-B Xu et al. (2022) | 97.1 | 90.5 | 82.4 | 77.5 | 96.6 | 96.1 | 93.6 | **88.6** | **97.9** | 86.9 | 90.3 | 62.8 | 88.4 |
| SSRT-B Sun et al. (2022) | **98.9** | 87.6 | **89.1** | **84.8** | 98.3 | **98.7** | **96.3** | 81.1 | 94.9 | 97.9 | 94.5 | 43.1 | 88.8 |
| NoisyTVT-B | 98.8 | **95.5** | 84.8 | 73.7 | **98.5** | 97.2 | 95.1 | 76.5 | 95.9 | **98.4** | **98.3** | **67.2** | **90.0** |

used UDA benchmarks. Here the positive noise is the linear transform noise identical to that used in the classification task. The positive noise is injected into the last layer of the model, the same as the classification task. The datasets include **Office Home** Venkateswara et al. (2017) and **VisDA2017** Peng et al. (2017). Detailed datasets introduction and experiments training settings are in the supplementary. The objective function is borrowed from TVT Yang et al. (2023a). The results are shown in Table 6 and 7 (the complete tables can be found in Tab. 21 and Tab. 22 in the Appendix). The NoisyTVT-B, i.e., TVT-B with positive noise, achieves better performance than existing works. These results demonstrate that positive noise also works in domain adaptation tasks, where out-of-distribution (OOD) data exists.

# 5 Conclusion

This study delves into the influence of entropy change on learning tasks, achieved by proactively introducing various types of noise into deep models. Our work conducts a comprehensive investigation into the impact of common noise types, such as Gaussian noise, linear transform noise, and salt-and-pepper noise, on deep learning models. Notably, we demonstrate that, under specific conditions, linear transform noise can positively affect deep models. The experimental results show that injecting positive noise into the latent space significantly enhances the prediction performance of deep models in image classification tasks, leading to new state-of-the-art results on ImageNet. These findings hold broad implications for future research and the potential development of more accurate models for improved real-world applications.

**Potential Impact**

The proposed theory exhibits versatility, opening avenues for exploring the application of positive noise across diverse deep learning tasks in computer vision and natural language processing. While our current findings and theoretical analysis are centered around addressing problems in the image classification tasks in computer vision, their implications extend far beyond. The optimal quality matrix derived from the linear transform noise suggests a potential for more effective model enhancement with larger datasets. This finding implies that the principles uncovered in our study may contribute to refining the strategies for improving large-scale language models trained on large-scale datasets. A potential concern is that those who possess high-quality large-scale datasets may primarily benefit from our research.

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

# A  Theoretical Foundations of Task Entropy

This section provides the theoretical foundations of task entropy, quantifying the complexity of learning tasks. The concept of task entropy was first proposed for the image level and formulated as Li (2022):

$$H(\mathcal{T}; \boldsymbol{X}) = - \sum_{\boldsymbol{Y} \in \mathcal{Y}} p(\boldsymbol{Y}|\boldsymbol{X}) \log p(\boldsymbol{Y}|\boldsymbol{X}) \tag{22}$$

The image $\boldsymbol{X}$ in the dataset are supposed to be independent of each other, as are the labels $\boldsymbol{Y}$. However, $\boldsymbol{X}$ and $\boldsymbol{Y}$ are not independent because of the correlation between a data sample $X$ and its corresponding label $Y$. Essentially, the task entropy is the entropy of $p(Y|X)$. Following the principle of task entropy, compelling evidence suggests that diminishing task complexity via reducing information entropy can enhance overall model performance Li (2022); Zhang et al. (2023).

Inspired by the concept of task entropy at the image level, we explore its extension to the latent space. The task entropy from the perspective of embeddings is defined as:

$$H(\mathcal{T}; \boldsymbol{Z}) \coloneqq H(\boldsymbol{Y}, \boldsymbol{Z}) - H(\boldsymbol{Z}) \tag{23}$$

where $\boldsymbol{Z}$ are the embeddings of the images $\boldsymbol{X}$. Here, we assume that the embedding $\boldsymbol{Z}$ and the vectorized label $\boldsymbol{Y}$ follow a multivariate normal distribution. We can transform the unknown distributions of $\boldsymbol{Z}$ and $\boldsymbol{Y}$ to approximately conform to normality by utilizing techniques such as Box-Cox transformation, log transform, etc., as described in Box & Cox (1964) and Feng et al. (2014) if they are not initially normal. After approximate transformation, the distribution of $\boldsymbol{Z}$ and $\boldsymbol{Y}$ can be expressed as:

$$\boldsymbol{Z} \sim \mathcal{N}(\boldsymbol{\mu_Z}, \boldsymbol{\Sigma_Z}), \boldsymbol{Y} \sim \mathcal{N}(\boldsymbol{\mu_Y}, \boldsymbol{\Sigma_Y}) \tag{24}$$

where

$$
\begin{aligned}
\boldsymbol{\mu_Z} &= \mathbb{E}[\boldsymbol{Z}] = (\mathbb{E}[Z_1], \mathbb{E}[Z_2], ..., \mathbb{E}[Z_k]])^T \\
\boldsymbol{\mu_Y} &= \mathbb{E}[\boldsymbol{Y}] = (\mathbb{E}[Y_1], \mathbb{E}[Y_2], ..., \mathbb{E}[Y_k]])^T \\
\boldsymbol{\Sigma_Z} &= \mathbb{E}[(\boldsymbol{Z} - \boldsymbol{\mu_Z})(\boldsymbol{Z} - \boldsymbol{\mu_Z})^T] \\
\boldsymbol{\Sigma_Y} &= \mathbb{E}[(\boldsymbol{Y} - \boldsymbol{\mu_Y})(\boldsymbol{Y} - \boldsymbol{\mu_Y})^T]
\end{aligned}
\tag{25}
$$

$k$ is the number of samples in the dataset, and $T$ represents the transpose of the matrix.

Then the conditional distribution of $\boldsymbol{Y}$ given $\boldsymbol{Z}$ is also normally distributed Mood (1950) Johnson et al. (1995), which can be formulated as:

$$\boldsymbol{Y}|\boldsymbol{Z} \sim \mathcal{N}(\mathbb{E}(\boldsymbol{Y}|\boldsymbol{Z} = Z), var(\boldsymbol{Y}|\boldsymbol{Z} = Z)) \tag{26}$$

where $\mathbb{E}(\boldsymbol{Y}|\boldsymbol{Z} = Z)$ is the mean of the label set $\boldsymbol{Y}$ given a sample $\boldsymbol{Z} = Z$ from the embeddings, and $var(\boldsymbol{Y}|\boldsymbol{Z} = Z)$ is the variance of $\boldsymbol{Y}$ given a sample from the embeddings. The conditional mean $\mathbb{E}[(\boldsymbol{Y}|\boldsymbol{Z} = Z)]$ and conditional variance $var(\boldsymbol{Y}|\boldsymbol{Z} = Z)$ can be calculated as:

$$\boldsymbol{\mu_{Y|Z=Z}} = \mathbb{E}[(\boldsymbol{Y}|\boldsymbol{Z} = Z)] = \boldsymbol{\mu_Y} + \boldsymbol{\Sigma_{YZ}}\boldsymbol{\Sigma_Z^{-1}}(Z - \boldsymbol{\mu_Z}) \tag{27}$$

$$\boldsymbol{\Sigma_{Y|Z=Z}} = var(\boldsymbol{Y}|\boldsymbol{Z} = Z) = \boldsymbol{\Sigma_Y} - \boldsymbol{\Sigma_{YX}}\boldsymbol{\Sigma_Z^{-1}}\boldsymbol{\Sigma_{ZY}} \tag{28}$$

where $\boldsymbol{\Sigma_{YZ}}$ and $\boldsymbol{\Sigma_{ZY}}$ are the cross-covariance matrices between $\boldsymbol{Y}$ and $\boldsymbol{Z}$, and between $\boldsymbol{Z}$ and $\boldsymbol{Y}$, respectively, and $\boldsymbol{\Sigma_Z^{-1}}$ denotes the inverse of the covariance matrix of $\boldsymbol{Z}$.

Now, we shall obtain the task entropy:

$$
\begin{aligned}
H(\mathcal{T}; \boldsymbol{Z}) =& -\sum_{\boldsymbol{Y} \in \mathcal{Y}} p(\boldsymbol{Y}|\boldsymbol{Z}) \log p(\boldsymbol{Y}|\boldsymbol{Z}) \\
=& -\mathbb{E}[\log p(\boldsymbol{Y}|\boldsymbol{Z})] \\
=& -\mathbb{E}\bigg[\log \big[(2\pi)^{-k/2}|\boldsymbol{\Sigma_Z}|^{-1/2} \\
& \times \exp\big(-\frac{1}{2}(\boldsymbol{Y}|\boldsymbol{Z} - \boldsymbol{\mu_{Y|Z}})^T \boldsymbol{\Sigma}_{\boldsymbol{Y|Z}}^{-1}(\boldsymbol{Y}|\boldsymbol{Z} - \boldsymbol{\mu_{Y|Z}}))\big]\bigg] \\
=& \frac{k}{2}\log(2\pi) + \frac{1}{2}\log|\boldsymbol{\Sigma_{Y|Z}}| \\
& + \frac{1}{2}\mathbb{E}[(\boldsymbol{Y}|\boldsymbol{Z} - \boldsymbol{\mu_{Y|Z}})^T \boldsymbol{\Sigma}_{\boldsymbol{Y|Z}}^{-1}(\boldsymbol{Y}|\boldsymbol{Z} - \boldsymbol{\mu_{Y|Z}})] \\
=& \frac{k}{2}(1 + \log(2\pi)) + \frac{1}{2}\log|\boldsymbol{\Sigma_{Y|Z}}|
\end{aligned}
\tag{29}
$$

where

$$
\begin{aligned}
\mathbb{E}\Big[(\boldsymbol{Y} \mid \boldsymbol{Z} - \boldsymbol{\mu_{Y|Z}})^T \Sigma_{\boldsymbol{Y|Z}}^{-1}(\boldsymbol{Y} \mid \boldsymbol{Z} - \boldsymbol{\mu_{Y|Z}})\Big] =& \mathbb{E}\Big[\mathrm{tr}\big((\boldsymbol{Y} \mid \boldsymbol{Z} - \boldsymbol{\mu_{Y|Z}})^T \Sigma_{\boldsymbol{Y|Z}}^{-1}(\boldsymbol{Y} \mid \boldsymbol{Z} - \boldsymbol{\mu_{Y|Z}})\big)\Big] \\
=& \mathbb{E}\Big[\mathrm{tr}\big(\Sigma_{\boldsymbol{Y|Z}}^{-1}(\boldsymbol{Y} \mid \boldsymbol{Z} - \boldsymbol{\mu_{Y|Z}})(\boldsymbol{Y} \mid \boldsymbol{Z} - \boldsymbol{\mu_{Y|Z}})^T\big)\Big] \\
=& \mathrm{tr}\Big(\Sigma_{\boldsymbol{Y|Z}}^{-1}\mathbb{E}\big[(\boldsymbol{Y} \mid \boldsymbol{Z} - \boldsymbol{\mu_{Y|Z}})(\boldsymbol{Y} \mid \boldsymbol{Z} - \boldsymbol{\mu_{Y|Z}})^T\big]\Big) \\
=& \mathrm{tr}\Big(\Sigma_{\boldsymbol{Y|Z}}^{-1}\Sigma_{\boldsymbol{Y|Z}}\Big) \\
=& \mathrm{tr}(\boldsymbol{I}_k) \\
=& k
\end{aligned}
\tag{30}
$$

Therefore, for a specific set of embeddings, we can find that the task entropy is only related to the variance of the $\boldsymbol{Y}|\boldsymbol{Z}$.

As we proactively inject different kinds of noises into the latent space, the task entropy with noise injection is defined as :

$$
\begin{cases}
H(\mathcal{T}; \boldsymbol{Z} + \boldsymbol{\epsilon}) \coloneqq H(\boldsymbol{Y}; \boldsymbol{Z} + \boldsymbol{\epsilon}) - H(\boldsymbol{Z}) & \boldsymbol{\epsilon} \text{ is additive noise} \\
H(\mathcal{T}; \boldsymbol{Z}\boldsymbol{\epsilon}) \coloneqq H(\boldsymbol{Y}; \boldsymbol{Z}\boldsymbol{\epsilon}) - H(\boldsymbol{Z}) & \boldsymbol{\epsilon} \text{ is multiplicative noise}
\end{cases}
\tag{31}
$$

Equation 31 diverges from the conventional definition of conditional entropy as our method introduces noise into the latent representations instead of the original images. The noises examined in this study are classified into additive and multiplicative categories. In the subsequent sections, we analyze the changes in task entropy resulting from the injection of common noises into the embeddings.

## B  The Impact of Gaussian Noise on Task Entropy

We begin by examining the impact of Gaussian noise on task entropy from the perspective of latent space.

### B.1  Inject Gaussian Noise in Latent Space

In this case, the task complexity is formulated as:

$$
H(\mathcal{T}; \boldsymbol{Z} + \boldsymbol{\epsilon}) = H(\boldsymbol{Y}; \boldsymbol{Z} + \boldsymbol{\epsilon}) - H(\boldsymbol{Z}).
\tag{32}
$$

Take advantage of the definition of task entropy, thus, the entropy change of injecting Gaussian noise in the latent space can be formulated as:

$$
\begin{aligned}
\triangle S(\mathcal{T}, \boldsymbol{\epsilon}) &= H(\mathcal{T}; \boldsymbol{Z}) - H(\mathcal{T}; \boldsymbol{Z} + \boldsymbol{\epsilon}) \\
&= H(\boldsymbol{Y}; \boldsymbol{Z}) - H(\boldsymbol{Z}) - (H(\boldsymbol{Y}; \boldsymbol{Z} + \boldsymbol{\epsilon}) - H(\boldsymbol{Z})) \\
&= H(\boldsymbol{Y}; \boldsymbol{Z}) - H(\boldsymbol{Y}; \boldsymbol{Z} + \boldsymbol{\epsilon}) \\
&= H(\boldsymbol{Y}|\boldsymbol{Z}) + H(\boldsymbol{Z}) - (H(\boldsymbol{Y}|\boldsymbol{Z} + \boldsymbol{\epsilon}) + H(\boldsymbol{Z} + \boldsymbol{\epsilon})) \\
&= \frac{1}{2} \log |\boldsymbol{\Sigma}_{\boldsymbol{Y}|\boldsymbol{Z}}| + \frac{1}{2} \log |\boldsymbol{\Sigma}_{\boldsymbol{Z}}| \\
&\quad - \frac{1}{2} \log |\boldsymbol{\Sigma}_{\boldsymbol{Y}|\boldsymbol{Z}+\boldsymbol{\epsilon}}| - \frac{1}{2} \log |\boldsymbol{\Sigma}_{\boldsymbol{Z}+\boldsymbol{\epsilon}}| \\
&= \frac{1}{2} \log \frac{|\boldsymbol{\Sigma}_{\boldsymbol{Z}}||\boldsymbol{\Sigma}_{\boldsymbol{Y}|\boldsymbol{Z}}|}{|\boldsymbol{\Sigma}_{\boldsymbol{Z}+\boldsymbol{\epsilon}}||\boldsymbol{\Sigma}_{\boldsymbol{Y}|\boldsymbol{Z}+\boldsymbol{\epsilon}}|} \\
&= \frac{1}{2} \log \frac{|\boldsymbol{\Sigma}_{\boldsymbol{Z}}||\boldsymbol{\Sigma}_{\boldsymbol{Y}} - \Sigma_{YZ}\Sigma_Z^{-1}\Sigma_{ZY}|}{|\boldsymbol{\Sigma}_{\boldsymbol{Z}+\boldsymbol{\epsilon}}||\boldsymbol{\Sigma}_{\boldsymbol{Y}} - \Sigma_{YZ}\Sigma_{Z+\epsilon}^{-1}\Sigma_{ZY}|}
\end{aligned}
\tag{33}
$$

where $\boldsymbol{\Sigma}_{\boldsymbol{Y}|\boldsymbol{Z}+\boldsymbol{\epsilon}} = \boldsymbol{\Sigma}_{\boldsymbol{Y}} - \boldsymbol{\Sigma}_{\boldsymbol{Y}(\boldsymbol{Z}+\boldsymbol{\epsilon})}\boldsymbol{\Sigma}_{\boldsymbol{Z}+\boldsymbol{\epsilon}}^{-1}\boldsymbol{\Sigma}_{(\boldsymbol{Z}+\boldsymbol{\epsilon})\boldsymbol{Y}}$. Since the Gaussian noise is independent of $\boldsymbol{Z}$ and $\boldsymbol{Y}$, we have $\boldsymbol{\Sigma}_{\boldsymbol{Y}(\boldsymbol{Z}+\boldsymbol{\epsilon})} = \boldsymbol{\Sigma}_{(\boldsymbol{Z}+\boldsymbol{\epsilon})\boldsymbol{Y}} = \boldsymbol{\Sigma}_{\boldsymbol{Y}\boldsymbol{Z}}$. The corresponding proof is:

$$
\begin{aligned}
\boldsymbol{\Sigma}_{(\boldsymbol{Z}+\boldsymbol{\epsilon})\boldsymbol{Y}} &= \mathbb{E}\left[(\boldsymbol{Z} + \boldsymbol{\epsilon}) - \mu_{\boldsymbol{Z}+\boldsymbol{\epsilon}}\right] \mathbb{E}\left[\boldsymbol{Y} - \mu_{\boldsymbol{Y}}\right] \\
&= \mathbb{E}\left[(\boldsymbol{Z} + \boldsymbol{\epsilon})\boldsymbol{Y}\right] - \mu_{\boldsymbol{Y}}\mathbb{E}\left[(\boldsymbol{Z} + \boldsymbol{\epsilon})\right] - \mu_{\boldsymbol{Z}+\boldsymbol{\epsilon}}\mathbb{E}\left[\boldsymbol{Y}\right] + \mu_{\boldsymbol{Y}}\mu_{\boldsymbol{Z}+\boldsymbol{\epsilon}} \\
&= \mathbb{E}\left[(\boldsymbol{Z} + \boldsymbol{\epsilon})\boldsymbol{Y}\right] - \mu_{\boldsymbol{Y}}\mathbb{E}\left[(\boldsymbol{Z} + \boldsymbol{\epsilon})\right] \\
&= \mathbb{E}\left[\boldsymbol{Z}\boldsymbol{Y}\right] + \mathbb{E}\left[\boldsymbol{\epsilon}\boldsymbol{Y}\right] - \mu_{\boldsymbol{Y}}\mu_{\boldsymbol{Z}} - \mu_{\boldsymbol{Y}}\mu_{\boldsymbol{\epsilon}} \\
&= \mathbb{E}\left[\boldsymbol{Z}\boldsymbol{Y}\right] - \mu_{\boldsymbol{Y}}\mu_{\boldsymbol{Z}} \\
&= \boldsymbol{\Sigma}_{\boldsymbol{Z}\boldsymbol{Y}}
\end{aligned}
\tag{34}
$$

Obviously,

$$
\begin{cases}
\triangle S(\mathcal{T}, \boldsymbol{\epsilon}) > 0 & if \quad \frac{|\boldsymbol{\Sigma}_{\boldsymbol{Z}}||\boldsymbol{\Sigma}_{\boldsymbol{Y}|\boldsymbol{Z}}|}{|\boldsymbol{\Sigma}_{\boldsymbol{Z}+\boldsymbol{\epsilon}}||\boldsymbol{\Sigma}_{\boldsymbol{Y}|\boldsymbol{Z}+\boldsymbol{\epsilon}}|} > 1 \\
\triangle S(\mathcal{T}, \boldsymbol{\epsilon}) \leq 0 & if \quad \frac{|\boldsymbol{\Sigma}_{\boldsymbol{Z}}||\boldsymbol{\Sigma}_{\boldsymbol{Y}|\boldsymbol{Z}}|}{|\boldsymbol{\Sigma}_{\boldsymbol{Z}+\boldsymbol{\epsilon}}||\boldsymbol{\Sigma}_{\boldsymbol{Y}|\boldsymbol{Z}+\boldsymbol{\epsilon}}|} \leq 1
\end{cases}
\tag{35}
$$

To find the relationship between $|\boldsymbol{\Sigma}_{\boldsymbol{Z}}||\boldsymbol{\Sigma}_{\boldsymbol{Y}|\boldsymbol{Z}}|$ and $|\boldsymbol{\Sigma}_{\boldsymbol{Z}+\boldsymbol{\epsilon}}||\boldsymbol{\Sigma}_{\boldsymbol{Y}|\boldsymbol{Z}+\boldsymbol{\epsilon}}|$, we need to determine the subterms in each of them. As we mentioned in the previous section, the embeddings of the images are independent of each other, and so are the labels.

$$
\begin{aligned}
\boldsymbol{\Sigma}_{\boldsymbol{Y}} &= \mathbb{E}[(\boldsymbol{Y} - \boldsymbol{\mu}_{\boldsymbol{Y}})(\boldsymbol{Y} - \boldsymbol{\mu}_{\boldsymbol{Y}})^T] \\
&= \mathbb{E}[\boldsymbol{Y}\boldsymbol{Y}^T] - \boldsymbol{\mu}_{\boldsymbol{Y}}\boldsymbol{\mu}_{\boldsymbol{Y}}^T \\
&= \mathrm{diag}(\sigma_{Y_1}^2, ..., \sigma_{Y_k}^2)
\end{aligned}
\tag{36}
$$

where

$$
\begin{cases}
\mathbb{E}\left[Y_i Y_j\right] - \mu_{Y_i}\mu_{Y_j} = 0, & i \neq j \\
\mathbb{E}\left[Y_i Y_j\right] - \mu_{Y_i}\mu_{Y_j} = \sigma_{Y_i}^2, & i = j
\end{cases}
\tag{37}
$$

The same procedure can be applied to $\boldsymbol{\Sigma}_{\boldsymbol{Y}(\boldsymbol{Z}+\boldsymbol{\epsilon})}$ and $\boldsymbol{\Sigma}_{\boldsymbol{Z}+\boldsymbol{\epsilon}}$. Therefore, We can obtain that $\boldsymbol{\Sigma}_{\boldsymbol{Y}} = \mathrm{diag}(\sigma_{Y_1}^2, ..., \sigma_{Y_k}^2)$,

$$
\boldsymbol{\Sigma}_{\boldsymbol{Y}(\boldsymbol{Z}+\boldsymbol{\epsilon})} = \mathrm{diag}(\mathrm{cov}(Y_1, Z_1 + \epsilon), ..., \mathrm{cov}(Y_k, Z_k + \epsilon))
\tag{38}
$$

and $\boldsymbol{\Sigma}_{\boldsymbol{Z}+\boldsymbol{\epsilon}}$ is:

$$
\boldsymbol{\Sigma}_{\boldsymbol{Z}+\boldsymbol{\epsilon}} =
\begin{bmatrix}
\sigma_{Z_1}^2 + \sigma_\epsilon^2 & \sigma_\epsilon^2 & \cdots & \sigma_\epsilon^2 & \sigma_\epsilon^2 \\
\sigma_\epsilon^2 & \sigma_{Z_2}^2 + \sigma_\epsilon^2 & \cdots & \sigma_\epsilon^2 & \sigma_\epsilon^2 \\
\vdots & \vdots & & \vdots & \vdots \\
\sigma_\epsilon^2 & \sigma_\epsilon^2 & \cdots & \sigma_{Z_{k-1}}^2 + \sigma_\epsilon^2 & \sigma_\epsilon^2 \\
\sigma_\epsilon^2 & \sigma_\epsilon^2 & \cdots & \sigma_\epsilon^2 & \sigma_{Z_k}^2 + \sigma_\epsilon^2
\end{bmatrix}
\tag{39}
$$
$$
= \mathrm{diag}(\sigma_{Z_1}^2, ..., \sigma_{Z_k}^2)\boldsymbol{I}_k + \sigma_\epsilon^2 \mathbf{1}_k
$$

where $\boldsymbol{I}_k$ is a $k \times k$ identity matrix and $\mathbf{1}_k$ is a all ones $k \times k$ matrix. We use $\boldsymbol{U}$ to represent $\mathrm{diag}(\sigma_{Z_1}^2, ..., \sigma_{Z_k}^2)\boldsymbol{I}_k$, and $\boldsymbol{u}$ to represent a all ones vector $[1, ..., 1]^T$. Thanks to the ShermanMorrison Formula Sherman & Morrison (1949) and Woodbury Formula Woodbury (1950), we can obtain the inverse of $\boldsymbol{\Sigma}_{\boldsymbol{Z}+\boldsymbol{\epsilon}}$ as:

$$
\begin{aligned}
\boldsymbol{\Sigma}_{\boldsymbol{Z}+\boldsymbol{\epsilon}}^{-1} &= (\boldsymbol{U} + \sigma_\epsilon^2 \boldsymbol{u}\boldsymbol{u}^T)^{-1} \\
&= \boldsymbol{U}^{-1} - \frac{\sigma_\epsilon^2}{1 + \sigma_\epsilon^2 \boldsymbol{u}^T \boldsymbol{U}^{-1}\boldsymbol{u}} \boldsymbol{U}^{-1}\boldsymbol{u}\boldsymbol{u}^T\boldsymbol{U}^{-1} \\
&= \boldsymbol{U}^{-1} - \frac{\sigma_\epsilon^2}{1 + \sum_{i=1}^k \frac{1}{\sigma_{Z_i}^2}} \boldsymbol{U}^{-1}\mathbf{1}_k\boldsymbol{U}^{-1} \\
&= \lambda
\begin{bmatrix}
\frac{1}{\lambda\sigma_{Z_1}^2} - \frac{1}{\sigma_{Z_1}^4} & -\frac{1}{\sigma_{Z_1}^2\sigma_{Z_2}^2} & \cdots & -\frac{1}{\sigma_{Z_1}^2\sigma_{Z_{k-1}}^2} & -\frac{1}{\sigma_{Z_1}^2\sigma_{Z_k}^2} \\
-\frac{1}{\sigma_{Z_2}^2\sigma_{Z_1}^2} & \frac{1}{\lambda\sigma_{Z_2}^2} - \frac{1}{\sigma_{Z_2}^4} & \cdots & -\frac{1}{\sigma_{Z_2}^2\sigma_{Z_{k-1}}^2} & -\frac{1}{\sigma_{Z_2}^2\sigma_{Z_k}^2} \\
\vdots & \vdots & & \vdots & \vdots \\
-\frac{1}{\sigma_{Z_{k-1}}^2\sigma_{Z_1}^2} & -\frac{1}{\sigma_{Z_{k-1}}^2\sigma_{Z_2}^2} & \cdots & \frac{1}{\lambda\sigma_{Z_{k-1}}^2} - \frac{1}{\sigma_{Z_{k-1}}^4} & -\frac{1}{\sigma_{Z_{k-1}}^2\sigma_{Z_k}^2} \\
-\frac{1}{\sigma_{Z_k}^2\sigma_{Z_1}^2} & -\frac{1}{\sigma_{Z_k}^2\sigma_{Z_2}^2} & \cdots & -\frac{1}{\sigma_{Z_k}^2\sigma_{Z_{k-1}}^2} & \frac{1}{\lambda\sigma_{Z_k}^2} - \frac{1}{\sigma_{Z_k}^4}
\end{bmatrix}
\end{aligned}
\tag{40}
$$

where $\boldsymbol{U}^{-1} = \mathrm{diag}((\sigma_{Z_1}^2)^{-1}, ..., (\sigma_{Z_k}^2)^{-1})$ and $\lambda = \frac{\sigma_\epsilon^2}{1 + \sum_{i=1}^k \frac{1}{\sigma_{Z_i}^2}}$.

Therefore, substitute Equation 40 into $|\boldsymbol{\Sigma}_{\boldsymbol{Y}} - \boldsymbol{\Sigma}_{\boldsymbol{Y}(\boldsymbol{Z}+\boldsymbol{\epsilon})}\boldsymbol{\Sigma}_{\boldsymbol{Z}+\boldsymbol{\epsilon}}^{-1}\boldsymbol{\Sigma}_{(\boldsymbol{Z}+\boldsymbol{\epsilon})\boldsymbol{Y}}|$, we can obtain:

$$
\begin{aligned}
&|\boldsymbol{\Sigma}_{\boldsymbol{Y}} - \boldsymbol{\Sigma}_{\boldsymbol{Y}(\boldsymbol{Z}+\boldsymbol{\epsilon})}\boldsymbol{\Sigma}_{\boldsymbol{Z}+\boldsymbol{\epsilon}}^{-1}\boldsymbol{\Sigma}_{(\boldsymbol{Z}+\boldsymbol{\epsilon})\boldsymbol{Y}}| \\
&= \left| 
\begin{bmatrix}
\sigma_{Y_1}^2 & \cdots & 0 \\
\vdots & \ddots & \vdots \\
0 & \cdots & \sigma_{Y_k}^2
\end{bmatrix}
-
\begin{bmatrix}
\mathrm{cov}(Y_1, Z_1+\epsilon) & \cdots & 0 \\
\vdots & \ddots & \vdots \\
0 & \cdots & \mathrm{cov}(Y_k, Z_k+\epsilon)
\end{bmatrix}
\boldsymbol{\Sigma}_{\boldsymbol{Z}+\boldsymbol{\epsilon}}^{-1}
\begin{bmatrix}
\mathrm{cov}(Y_1, Z_1+\epsilon) & \cdots & 0 \\
\vdots & \ddots & \vdots \\
0 & \cdots & \mathrm{cov}(Y_k, Z_k+\epsilon)
\end{bmatrix}
\right| \\
&= \left|
\begin{bmatrix}
\sigma_{Y_1}^2 - \mathrm{cov}^2(Y_1, Z_1+\epsilon)(\frac{1}{\sigma_{Z_1}^2} - \frac{\lambda}{\sigma_{Z_1}^4}) & \cdots & \mathrm{cov}(Y_1, Z_1+\epsilon)\mathrm{cov}(Y_k, Z_k+\epsilon)\frac{\lambda}{\sigma_{Z_1}^2\sigma_{Z_k}^2} \\
\vdots & & \vdots \\
\mathrm{cov}(Y_k, Z_k+\epsilon)\mathrm{cov}(Y_1, Z_1+\epsilon)\frac{\lambda}{\sigma_{Z_k}^2\sigma_{Z_1}^2} & \cdots & \sigma_{Y_k}^2 - \mathrm{cov}^2(Y_k, Z_k+\epsilon)(\frac{1}{\sigma_{Z_k}^2} - \frac{\lambda}{\sigma_{Z_k}^4})
\end{bmatrix}
\right| \\
&= \left|
\begin{bmatrix}
\sigma_{Y_1}^2 - \frac{1}{\sigma_{Z_1}^2}\mathrm{cov}^2(Y_1, Z_1) & & \\
& \ddots & \\
& & \sigma_{Y_k}^2 - \frac{1}{\sigma_{Z_k}^2}\mathrm{cov}^2(Y_k, Z_k)
\end{bmatrix}
+ \lambda
\begin{bmatrix}
\frac{1}{\sigma_{Z_1}^4}\mathrm{cov}^2(Y_1, Z_1) & \cdots & \frac{1}{\sigma_{Z_1}^2\sigma_{Z_k}^2}\mathrm{cov}(Y_1, Z_1)\mathrm{cov}(Y_k, Z_k) \\
\vdots & & \vdots \\
\frac{1}{\sigma_{Z_k}^2\sigma_{Z_1}^2}\mathrm{cov}(Y_k, Z_k)\mathrm{cov}(Y_1, Z_1) & \cdots & \frac{1}{\sigma_{Z_k}^4}\mathrm{cov}^2(Y_k, Z_k)
\end{bmatrix}
\right|
\end{aligned}
\tag{41}
$$

We use the notation $\boldsymbol{v} = \left[ \frac{1}{\sigma_{Z_1}^2}\mathrm{cov}(Y_1, Z_1) \quad \cdots \quad \frac{1}{\sigma_{Z_k}^2}\mathrm{cov}(Y_k, Z_k) \right]^T$, and $\boldsymbol{V} = \mathrm{diag}(\frac{1}{\sigma_{Z_1}^2}\mathrm{cov}^2(Y_1, Z_1), \cdots, \frac{1}{\sigma_{Z_k}^2}\mathrm{cov}^2(Y_k, Z_k))$. And utilize the rule of determinants of sums Marcus (1990), then we have:

$$
\begin{aligned}
\left| \boldsymbol{\Sigma}_{\boldsymbol{Y}} - \boldsymbol{\Sigma}_{\boldsymbol{Y}(\boldsymbol{Z}+\boldsymbol{\epsilon})}\boldsymbol{\Sigma}_{\boldsymbol{Z}+\boldsymbol{\epsilon}}^{-1}\boldsymbol{\Sigma}_{(\boldsymbol{Z}+\boldsymbol{\epsilon})\boldsymbol{Y}} \right| &= \left| (\boldsymbol{\Sigma}_{\boldsymbol{Y}} - \boldsymbol{V}) + \lambda \boldsymbol{v}\boldsymbol{v}^T \right| \\
&= |\boldsymbol{\Sigma}_{\boldsymbol{Y}} - \boldsymbol{V}| + \lambda \boldsymbol{v}^T(\boldsymbol{\Sigma}_{\boldsymbol{Y}} - \boldsymbol{V})^{-1}\boldsymbol{v}.
\end{aligned}
\tag{42}
$$

where $(\mathbf{\Sigma_Y} - \mathbf{V})^*$ is the adjoint of the matrix $(\mathbf{\Sigma_Y} - \mathbf{V})$. For simplicity, we can rewrite $|\mathbf{\Sigma_Y} - \mathbf{\Sigma_{Y(Z+\epsilon)}}\mathbf{\Sigma_{Z+\epsilon}^{-1}}\mathbf{\Sigma_{(Z+\epsilon)Y}}|$ as:

$$
\begin{aligned}
&\left|\mathbf{\Sigma_Y} - \mathbf{\Sigma_{Y(Z+\epsilon)}}\mathbf{\Sigma_{Z+\epsilon}^{-1}}\mathbf{\Sigma_{(Z+\epsilon)Y}}\right| \\
&= \prod_{i=1}^{k}\left(\sigma_{Y_i}^2 - \mathrm{cov}^2(Y_i, Z_i) \cdot \frac{1}{\sigma_{Z_i}^2}\right) + \Omega
\end{aligned}
\tag{43}
$$

where $\Omega = \lambda \cdot \mathbf{v}^T \cdot (\mathbf{\Sigma_Y} - \mathbf{V})^* \cdot \mathbf{v}$. The specific value of $\Omega$ can be obtained as:

$$
\Omega = \lambda \cdot \begin{bmatrix} \frac{1}{\sigma_{Z_1}^2}\mathrm{cov}(Y_1, Z_1) & \cdots & \frac{1}{\sigma_{Z_k}^2}\mathrm{cov}(Y_k, Z_k) \end{bmatrix} \cdot \begin{bmatrix} V_{11} & 0 & 0 \\ 0 & \ddots & 0 \\ 0 & 0 & V_{kk} \end{bmatrix} \cdot \begin{bmatrix} \frac{1}{\sigma_{Z_1}^2}\mathrm{cov}(Y_1, Z_1) \\ \vdots \\ \frac{1}{\sigma_{Z_k}^2}\mathrm{cov}(Y_k, Z_k) \end{bmatrix}
\tag{44}
$$

where the elements $V_{ii}, i \in [1, k]$ are minors of the matrix and expressed as:

$$
V_{ii} = \prod_{j=1, j\neq i}^{k} \left[\sigma_{Y_j}^2 - \frac{1}{\sigma_{Z_j}^2}\mathrm{cov}^2(Z_j, Y_j)\right]
\tag{45}
$$

After some necessary steps, Equation 44 is reduced to:

$$
\begin{aligned}
\Omega &= \lambda \sum_{i=1}^{k} \frac{\frac{1}{\sigma_{Z_i}^4}\mathrm{cov}^2(Y_i, Z_i)\prod_{j=1}^{k}(\sigma_{Y_j}^2 - \mathrm{cov}^2(Y_j, Z_j)\frac{1}{\sigma_{Z_j}^2})}{(\sigma_{Y_i}^2 - \mathrm{cov}^2(Y_i, Z_i)\frac{1}{\sigma_{Z_i}^2})} \\
&= \lambda \prod_{i=1}^{k}(\sigma_{Y_i}^2 - \mathrm{cov}^2(Y_i, Z_i)\frac{1}{\sigma_{Z_i}^2}) \cdot \sum_{i=1}^{k} \frac{\mathrm{cov}^2(Z_i, Y_i)}{\sigma_{Z_i}^2(\sigma_{Z_i}^2\sigma_{Y_i}^2 - \mathrm{cov}^2(Z_i, Y_i))}
\end{aligned}
\tag{46}
$$

Substitute Equation 46 into Equation 43, we can get:

$$
\begin{aligned}
&\left|\mathbf{\Sigma_Y} - \mathbf{\Sigma_{Y(Z+\epsilon)}}\mathbf{\Sigma_{Z+\epsilon}^{-1}}\mathbf{\Sigma_{(Z+\epsilon)Y}}\right| \\
&= \prod_{i=1}^{k}(\sigma_{Y_i}^2 - \mathrm{cov}^2(Y_i, Z_i)\frac{1}{\sigma_{Z_i}^2}) \cdot (1 + \lambda\sum_{i=1}^{k}\frac{\mathrm{cov}^2(Z_i, Y_i)}{\sigma_{Z_i}^2(\sigma_{Z_i}^2\sigma_{Y_i}^2 - \mathrm{cov}^2(Z_i, Y_i))})
\end{aligned}
\tag{47}
$$

Accordingly, $|\mathbf{\Sigma_Y} - \mathbf{\Sigma_{YZ}}\mathbf{\Sigma_Z^{-1}}\mathbf{\Sigma_{ZY}}|$ is:

$$
|\mathbf{\Sigma_Y} - \mathbf{\Sigma_{YZ}}\mathbf{\Sigma_Z^{-1}}\mathbf{\Sigma_{ZY}}| = \prod_{i=1}^{k}(\sigma_{Y_i}^2 - \frac{1}{\sigma_{Z_i}^2}\mathrm{cov}^2(Z_i, Y_i))
\tag{48}
$$

As a result, $\frac{|\mathbf{\Sigma_{Y|Z+\epsilon}}|}{|\mathbf{\Sigma_{Y|Z}}|}$ is expressed as:

$$
\frac{|\mathbf{\Sigma_{Y|Z}}|}{|\mathbf{\Sigma_{Y|Z+\epsilon}}|} = \frac{\prod_{i=1}^{k}(\sigma_{Y_i}^2 - \frac{1}{\sigma_{Z_i}^2}\mathrm{cov}^2(Z_i, Y_i))}{\prod_{i=1}^{k}(\sigma_{Y_i}^2 - \mathrm{cov}^2(Y_i, Z_i)\frac{1}{\sigma_{Z_i}^2}) \cdot (1 + \lambda\sum_{i=1}^{k}\frac{\mathrm{cov}^2(Z_i, Y_i)}{\sigma_{Z_i}^2(\sigma_{Z_i}^2\sigma_{Y_i}^2 - \mathrm{cov}^2(Z_i, Y_i))})}
\tag{49}
$$

Combine Equations 49 and 39 together, the entropy change is expressed as:

$$
\triangle S(\mathcal{T}, \boldsymbol{\epsilon}) = \frac{1}{2}\log\frac{1}{(1 + \sigma_\epsilon^2\sum_{i=1}^{k}\frac{1}{\sigma_{Z_i}^2})(1 + \lambda\sum_{i=1}^{k}\frac{\mathrm{cov}^2(Z_i, Y_i)}{\sigma_{Z_i}^2(\sigma_{Z_i}^2\sigma_{Y_i}^2 - \mathrm{cov}^2(Z_i, Y_i))})}
\tag{50}
$$

It is difficult to tell that Equation 50 is greater or smaller than 0 directly. But one thing for sure is that when there is no Gaussian noise, Equation 50 equals 0. However, we can use another way to compare the

numerator and denominator in Equation 50. Instead, we use the symbol $M$ to compare the numerator and denominator using subtraction. Let:

$$M = 1 - (1 + \sigma_\epsilon^2 \sum_{i=1}^k \frac{1}{\sigma_{Z_i}^2})(1 + \lambda \sum_{i=1}^k \frac{\text{cov}^2(Z_i, Y_i)}{\sigma_{Z_i}^2(\sigma_{Z_i}^2 \sigma_{Y_i}^2 - \text{cov}^2(Z_i, Y_i))}) \tag{51}$$

Obviously, the variance $\sigma_\epsilon^2$ of the Gaussian noise control the result of $M$, while the mean $\mu_\epsilon$ has no influence. When $\sigma_\epsilon$ approaching 0, we have:

$$\lim_{\sigma_\epsilon^2 \to 0} M = 0 \tag{52}$$

To determine if Gaussian noise can be positive noise, we need to determine whether the entropy change is large or smaller than 0.

$$\begin{cases} \triangle S(\mathcal{T}, \epsilon) > 0 & \text{if } M > 0 \\ \triangle S(\mathcal{T}, \epsilon) \leq 0 & \text{if } M \leq 0 \end{cases} \tag{53}$$

From the above equations, the sign of the entropy change is determined by the statistical properties of the embeddings and labels. Since $\epsilon^2 \geq 0$, $\lambda \geq 0$ and $\sum_{i=1}^k \frac{1}{\sigma_{Z_i}^2} \geq 0$, we need to have a deep dive into the residual part, i.e.,

$$\sum_{i=1}^k \frac{\text{cov}^2(Z_i, Y_i)}{\sigma_{Z_i}^2(\sigma_{Z_i}^2 \sigma_{Y_i}^2 - \text{cov}^2(Z_i, Y_i))} = \sum_{i=1}^k \frac{\text{cov}^2(Z_i, Y_i)}{\sigma_{Z_i}^4 \sigma_{Y_i}^2(1 - \rho_{Z_i Y_i}^2)} \tag{54}$$

where $\rho_{Z_i Y_i}$ is the correlation coefficient, and $\rho_{Z_i Y_i}^2 \in [0, 1]$. Eq. 54 is greater than 0, As a result, the sign of the entropy change in the Gaussian noise case is negative. We can conclude that Gaussian noise added to the latent space is harmful to the task.

## B.2 Add Gaussian Noise to Raw Images

Assuming that the pixels of the raw images follow a Gaussian distribution. The variation of task complexity by adding Gaussian noise to raw images can be formulated as:

$$\begin{aligned} \triangle S(\mathcal{T}, \epsilon) &= H(\mathcal{T}; \boldsymbol{X}) - H(\mathcal{T}; \boldsymbol{X} + \boldsymbol{\epsilon}) \\ &= \frac{1}{2} \log |\boldsymbol{\Sigma}_{\boldsymbol{Y}|\boldsymbol{X}}| - \frac{1}{2} \log |\boldsymbol{\Sigma}_{\boldsymbol{Y}|\boldsymbol{X}+\boldsymbol{\epsilon}}| \\ &= \frac{1}{2} \log \frac{|\boldsymbol{\Sigma}_{\boldsymbol{Y}|\boldsymbol{X}}|}{|\boldsymbol{\Sigma}_{\boldsymbol{Y}|\boldsymbol{X}+\boldsymbol{\epsilon}}|} \\ &= \frac{1}{2} \log \frac{|\boldsymbol{\Sigma}_{\boldsymbol{Y}} - \boldsymbol{\Sigma}_{\boldsymbol{YX}} \boldsymbol{\Sigma}_{\boldsymbol{X}}^{-1} \boldsymbol{\Sigma}_{\boldsymbol{XY}}|}{|\boldsymbol{\Sigma}_{\boldsymbol{Y}} - \boldsymbol{\Sigma}_{\boldsymbol{Y}(\boldsymbol{X}+\boldsymbol{\epsilon})} \boldsymbol{\Sigma}_{\boldsymbol{X}+\boldsymbol{\epsilon}}^{-1} \boldsymbol{\Sigma}_{(\boldsymbol{X}+\boldsymbol{\epsilon})\boldsymbol{Y}}|} \\ &= \frac{1}{2} \log \frac{|\boldsymbol{\Sigma}_{\boldsymbol{Y}} - \boldsymbol{\Sigma}_{\boldsymbol{YX}} \boldsymbol{\Sigma}_{\boldsymbol{X}}^{-1} \boldsymbol{\Sigma}_{\boldsymbol{XY}}|}{|\boldsymbol{\Sigma}_{\boldsymbol{Y}} - \boldsymbol{\Sigma}_{\boldsymbol{YX}} \boldsymbol{\Sigma}_{\boldsymbol{X}+\boldsymbol{\epsilon}}^{-1} \boldsymbol{\Sigma}_{\boldsymbol{XY}}|} \end{aligned} \tag{55}$$

Borrow the equations from the case of Gaussian noise added to the latent space, we have:

$$\triangle S(\mathcal{T}, \epsilon) = \frac{1}{2} \log \frac{1}{1 + \lambda \sum_{i=1}^k \frac{\text{cov}^2(X_i, Y_i)}{\sigma_{X_i}^2(\sigma_{X_i}^2 \sigma_{Y_i}^2 - \text{cov}^2(X_i, Y_i))}} \tag{56}$$

Clearly, the introduction of Gaussian noise to each pixel in the original images has a detrimental impact on the task. **Note** that some studies have empirically shown that adding Gaussian noise to partial pixels of input images may be beneficial to the learning task Li (2022) Zhang et al. (2023).

## C Impact of Linear Transform Noise on Task Entropy

In our work, concerning the image level perspective, "linear transform noise" denotes an image that is perturbed by another image or a combination of other images. From the viewpoint of embeddings, "linear transform noise" refers to an embedding perturbed by another embedding or the combination of other embeddings.

### C.1 Inject Linear Transform Noise in Latent Space

The entropy change of injecting linear transform noise into embeddings can be formulated as:

$$
\begin{aligned}
\triangle S(\mathcal{T}, Q\mathbf{Z}) =& H(\mathcal{T}; \mathbf{Z}) - H(\mathcal{T}; \mathbf{Z} + Q\mathbf{Z}) \\
=& H(\mathbf{Y}; \mathbf{Z}) - H(\mathbf{Z}) - (H(\mathbf{Y}; \mathbf{Z} + Q\mathbf{Z}) - H(\mathbf{Z})) \\
=& H(\mathbf{Y}; \mathbf{Z}) - H(\mathbf{Y}; \mathbf{Z} + Q\mathbf{Z}) \\
=& \frac{1}{2} \log \frac{|\mathbf{\Sigma}_{\mathbf{Z}}||\mathbf{\Sigma}_{\mathbf{Y}} - \mathbf{\Sigma}_{\mathbf{YZ}}\mathbf{\Sigma}_{\mathbf{Z}}^{-1}\mathbf{\Sigma}_{\mathbf{ZY}}|}{|\mathbf{\Sigma}_{(I+Q)\mathbf{Z}}||\mathbf{\Sigma}_{\mathbf{Y}} - \mathbf{\Sigma}_{\mathbf{YZ}}\mathbf{\Sigma}_{\mathbf{Z}}^{-1}\mathbf{\Sigma}_{\mathbf{ZY}}|} \\
=& \frac{1}{2} \log \frac{1}{|I+Q|^2} \\
=& -\log |I+Q|
\end{aligned}
\tag{57}
$$

Since we want the entropy change to be greater than 0, we can formulate Equation 57 as an optimization problem:

$$
\begin{aligned}
\max_Q &\ \triangle S(\mathcal{T}, Q\mathbf{Z}) \\
s.t. &\ rank(I+Q) = k \\
&\ Q \sim I \\
&\ [I+Q]_{ii} \geq [I+Q]_{ij}, i \neq j \\
&\ \|[I+Q]_i\|_1 = 1
\end{aligned}
\tag{58}
$$

where $\sim$ means the row equivalence. The key to determining whether the linear transform is positive noise or not lies in the matrix of $Q$. The most important step is to ensure that $I + Q$ is invertible, which is $|(I+Q)| \neq 0$. For this, we need to investigate what leads $I + Q$ to be rank-deficient. The third constraint is to make the trained classifier get enough information about a specific embedding of an image and correctly predict the corresponding label. For instance, when an embedding $Z_1$ is perturbed by another embedding $Z_2$, the classifier predominantly relies on the information from $Z_1$ to predict the label $Y_1$. Conversely, if the perturbed embedding $Z_2$ takes precedence, the classifier struggles to accurately predict the label $Y_1$ and is more likely to predict it as label $Y_2$. The fourth constraint is the normalization of latent representations.

**Rank Deficiency Cases.** To avoid causing a rank deficiency of $I + Q$, we need to figure out the conditions that lead to rank deficiency. Here we show a simple case causing the rank deficiency. When the matrix $Q$ is a backward identity matrix Horn & R. (2012),

$$
Q_{i,j} = \begin{cases} 1, & i+j = k+1 \\ 0, & i+j \neq k+1 \end{cases}
\tag{59}
$$

i.e.,

$$
Q = \begin{bmatrix}
0 & 0 & ... & 0 & 0 & 1 \\
0 & 0 & ... & 0 & 1 & 0 \\
\vdots & \vdots & & \vdots & \vdots & \vdots \\
0 & 1 & ... & 0 & 0 & 0 \\
1 & 0 & ... & 0 & 0 & 0
\end{bmatrix}
\tag{60}
$$

then $(I+Q)$ will be:

$$
I + Q = \begin{bmatrix}
1 & 0 & ... & 0 & 0 & 1 \\
0 & 1 & ... & 0 & 1 & 0 \\
\vdots & \vdots & & \vdots & \vdots & \vdots \\
0 & 1 & ... & 0 & 1 & 0 \\
1 & 0 & ... & 0 & 0 & 1
\end{bmatrix}
\tag{61}
$$

Thus, $I + Q$ will be rank-deficient when $Q$ is a backward identity. In fact, when the following constraints are satisfied, the $I + Q$ will be rank-deficient:

$$\text{HermiteForm}(I + Q)_i = \mathbf{0}, \quad \exists i \in [1, k] \tag{62}$$

where index $i$ is the row index, in this paper, the row index starts from 1, and HermiteForm is the Hermite normal form Kannan & Bachem (1979).

**Full Rank Cases.** Except for the rank deficiency cases, $I + Q$ has full rank and is invertible. Since $Q$ is a row equivalent to the identity matrix, we need to introduce the three types of elementary row operations as follows Shores (2007).

▷ 1 **Row Swap** Exchange rows.
Row swap here allows exchanging any number of rows. This is slightly different from the original one that only allows any two-row exchange since following the original row swap will lead to a rank deficiency. When the $Q$ is derived from $I$ with **Row Swap**, it will break the third constraint. Therefore, **Row Swap** merely is considered harmful and would degrade the performance of deep models.

▷ 2 **Scalar Multiplication** Multiply any row by a constant $\beta$. This breaks the fourth constraint, thus degrading the performance of deep models.

▷ 3 **Row Sum** Add a multiple of one row to another row. Then the matrix $I + Q$ would be like:

$$I + Q = \begin{bmatrix} 1 & & & \\ & . & & \\ & & . & \\ & & & . & \\ & & & & 1 \end{bmatrix} + \begin{bmatrix} 1 & & & \\ & . & & \beta \\ & & . & \\ & & & . & \\ & & & & 1 \end{bmatrix}$$
$$= \begin{bmatrix} 2 & & & \\ & . & \beta \\ & & . & \\ & & & . \\ & & & & 2 \end{bmatrix} \tag{63}$$

where $\beta$ can be at a random position beside the diagonal. As we can see from the simple example, **Row Sum** breaks the fourth constraint and makes entropy change smaller than 0.

From the above discussion, none of the single elementary row operations can guarantee positive effects on deep models.

However, if we combine the elementary row operations, it is possible to make the entropy change greater than 0 as well as satisfy the constraints. For example, we combine the **Row Sum** and **Scalar Multiplication** to generate the $Q$:

$$I + Q = \begin{bmatrix} 1 & & & \\ & . & & \\ & & . & \\ & & & . \\ & & & & 1 \end{bmatrix} + \begin{bmatrix} -0.5 & 0.5 & & \\ & . & . & \\ & & . & . & \\ & & & . & 0.5 \\ 0.5 & & & & -0.5 \end{bmatrix}$$
$$= \begin{bmatrix} 0.5 & 0.5 & & \\ & . & . & \\ & & . & . \\ & & & . & 0.5 \\ 0.5 & & & & 0.5 \end{bmatrix} \tag{64}$$

In this case, $\triangle S(\mathcal{T}, Q\boldsymbol{X}) > 0$ when $Q = -0.5I$. The constraints are satisfied. This is just a simple case of adding linear transform noise that benefits deep models. Actually, there exists a design space of $Q$ that

within the design space, deep models can reduce task entropy by injecting linear transform noise into the embeddings. To this end, we demonstrate that linear transform can be positive noise.

From the discussion in this section, we can draw conclusions that **Linear Transform Noise** can be positive under certain conditions, while **Gaussian Noise** and **Salt-and-pepper Noise** are harmful noise. From the above analysis, the conditions that satisfy positive noise form a design space. Exploring the design space of positive noise is an important topic for future work.

### C.1.1  Optimal Quality Matrix of Linear Transform Noise

The optimal quality matrix should maximize the entropy change and therefore theoretically define the minimized task complexity. The optimization problem as formulated in Equation 15 is:

$$
\begin{aligned}
\max_{Q} &\ -\log |I + Q| \\
s.t.\ &\ rank(I + Q) = k \\
&\ Q \sim I \\
&\ [I + Q]_{ii} \geq [I + Q]_{ij}, i \neq j \\
&\ \|[I + Q]_i\|_1 = 1
\end{aligned}
\tag{65}
$$

Maximizing the entropy change is to minimize the determinant of the matrix sum of $I$ and $Q$. A simple but straight way is to design the matrix $Q$ that makes the elements in $I + Q$ equal, i.e.,

$$
I + Q = \begin{bmatrix} 1/k & \cdots & 1/k \\ \vdots & \ddots & \vdots \\ 1/k & \cdots & 1/k \end{bmatrix}
\tag{66}
$$

The determinant of the above equation is 0, but it breaks the first constraint of $rank(I + Q) = k$. However, by adding a small constant into the diagonal, and minus another constant by other elements, we can get:

$$
I + Q = \begin{bmatrix} 1/k + c_1 & \cdots & & 1/k - c_2 \\ 1/k - c_2 & \ddots & & \vdots \\ \vdots & & \ddots & 1/k - c_2 \\ 1/k - c_2 & \cdots & 1/k - c_2 & 1/k + c_1 \end{bmatrix}
\tag{67}
$$

Under the constraints, we can obtain the two constants that fulfill the requirements:

$$
c_1 = \frac{k-1}{k(k+1)}, \quad c_2 = \frac{1}{k(k+1)}
\tag{68}
$$

Therefore, the corresponding $Q$ is:

$$
Q_{optimal} = \mathrm{diag}\left(\frac{1}{k+1} - 1, \ldots, \frac{1}{k+1} - 1\right) + \frac{1}{k+1}\mathbf{1}_{k \times k}
\tag{69}
$$

and the corresponding $I + Q$ is:

$$
I + Q = \begin{bmatrix} 2/(k+1) & \cdots & & 1/(k+1) \\ 1/(k+1) & \ddots & & \vdots \\ \vdots & & \ddots & 1/(k+1) \\ 1/(k+1) & \cdots & 1/(k+1) & 2/(k+1) \end{bmatrix}
\tag{70}
$$

As a result, the determinant of optimal $I + Q$ can be obtained by following the identical procedure as Equation 42:

$$
|I + Q| = \frac{1}{(k+1)^{k-1}}
\tag{71}
$$

The upper boundary of entropy change of linear transform noise is determined:

$$
\triangle S(\mathcal{T}, Q\boldsymbol{X})_{upper} = (k-1)\log(k+1)
\tag{72}
$$

### C.2 Add Linear Transform Noise to Raw Images

In this case, the task entropy with linear transform noise can be formulated as:

$$
\begin{aligned}
H(\mathcal{T}; \boldsymbol{X} + Q\boldsymbol{X}) &= - \sum_{\boldsymbol{Y} \in \mathcal{Y}} p(\boldsymbol{Y}|\boldsymbol{X} + Q\boldsymbol{X}) \log p(\boldsymbol{Y}|\boldsymbol{X} + Q\boldsymbol{X}) \\
&= - \sum_{\boldsymbol{Y} \in \mathcal{Y}} p(\boldsymbol{Y}|(I + Q)\boldsymbol{X}) \log p(\boldsymbol{Y}|(I + Q)\boldsymbol{X})
\end{aligned}
\tag{73}
$$

where $I$ is an identity matrix, and $Q$ is derived from $I$ using elementary row operations. Assuming that the pixels of the raw images follow a Gaussian distribution. The conditional distribution of $\boldsymbol{Y}$ given $\boldsymbol{X} + Q\boldsymbol{X}$ is also multivariate subjected to the normal distribution, which can be formulated as:

$$
\boldsymbol{Y}|(I + Q)\boldsymbol{X} \sim \mathcal{N}(\mathbb{E}(\boldsymbol{Y}|(I + Q)\boldsymbol{X}), var(\boldsymbol{Y}|(I + Q)\boldsymbol{X}))
\tag{74}
$$

Since the linear transform matrix is invertible, applying the linear transform to $\boldsymbol{X}$ does not alter the distribution of the $\boldsymbol{X}$. It is straightforward to obtain:

$$
\boldsymbol{\mu_{Y|(I+Q)X}} = \boldsymbol{\mu_Y} + \boldsymbol{\Sigma_{YX}\Sigma_X^{-1}}(I + Q)^{-1}((I + Q)X - (I + Q)\boldsymbol{\mu_X})
\tag{75}
$$

$$
\boldsymbol{\Sigma_{(Y|(I+Q)X)}} = \boldsymbol{\Sigma_Y} - \boldsymbol{\Sigma_{YX}\Sigma_X^{-1}\Sigma_{XY}}
\tag{76}
$$

Thus, the variation of task entropy adding linear transform noise can be formulated as:

$$
\begin{aligned}
\triangle S(\mathcal{T}, Q\boldsymbol{X}) &= H(\mathcal{T}; \boldsymbol{X}) - H(\mathcal{T}; \boldsymbol{X} + Q\boldsymbol{X}) \\
&= \frac{1}{2} \log |\boldsymbol{\Sigma_{Y|X}}| - \frac{1}{2} \log |\Sigma_{\boldsymbol{Y}|\boldsymbol{X}+Q\boldsymbol{X}}| \\
&= \frac{1}{2} \log \frac{|\boldsymbol{\Sigma_{Y|X}}|}{|\Sigma_{\boldsymbol{Y}|\boldsymbol{X}+Q\boldsymbol{X}}|} \\
&= \frac{1}{2} \log \frac{|\boldsymbol{\Sigma_Y} - \boldsymbol{\Sigma_{YX}\Sigma_X^{-1}\Sigma_{XY}}|}{|\boldsymbol{\Sigma_Y} - \boldsymbol{\Sigma_{YX}\Sigma_X^{-1}\Sigma_{XY}}|} \\
&= 0
\end{aligned}
\tag{77}
$$

The entropy change of 0 indicates that the implementation of linear transformation to the raw images could not help reduce the complexity of the task.

## D  Influence of Salt-and-pepper Noise on Task Entropy

Salt-and-pepper noise is a common type of noise that can occur in images due to various factors, such as signal transmission errors, faulty sensors, or other environmental factors Chan et al. (2005). Salt-and-pepper noise is often considered to be an independent process because it is a type of random noise that affects individual pixels in an image independently of each other Gonzales & Wintz (1987).

### D.1 Inject Salt-and-pepper Noise in Latent Space

The entropy change of injecting salt-and-pepper noise can be formulated as:

$$
\begin{aligned}
\triangle S(\mathcal{T}, Q\boldsymbol{Z}) =& H(\mathcal{T}; \boldsymbol{Z}) - H(\mathcal{T}; \boldsymbol{Z}\boldsymbol{\epsilon}) \\
=& H(\boldsymbol{Y}; \boldsymbol{Z}) - H(\boldsymbol{Z}) - (H(\boldsymbol{Y}; \boldsymbol{Z}\boldsymbol{\epsilon}) - H(\boldsymbol{Z})) \\
=& H(\boldsymbol{Y}; \boldsymbol{Z}) - H(\boldsymbol{Y}; \boldsymbol{Z}\boldsymbol{\epsilon}) \\
=& -\sum_{\boldsymbol{Z}\in\mathcal{Z}}\sum_{\boldsymbol{Y}\in\mathcal{Y}} p(\boldsymbol{Z}, \boldsymbol{Y}) \log p(\boldsymbol{Z}, \boldsymbol{Y}) + \sum_{\boldsymbol{Z}\in\mathcal{Z}}\sum_{\boldsymbol{Y}\in\mathcal{Y}}\sum_{\boldsymbol{\epsilon}\in\mathcal{E}} p(\boldsymbol{Z}\boldsymbol{\epsilon}, \boldsymbol{Y}) \log p(\boldsymbol{Z}\boldsymbol{\epsilon}, \boldsymbol{Y}) \\
=& \mathbb{E}\left[\log \frac{1}{p(\boldsymbol{Z}, \boldsymbol{Y})}\right] - \mathbb{E}\left[\log \frac{1}{p(\boldsymbol{Z}\boldsymbol{\epsilon}, \boldsymbol{Y})}\right] \\
=& \mathbb{E}\left[\log \frac{1}{p(\boldsymbol{Z}, \boldsymbol{Y})}\right] - \mathbb{E}\left[\log \frac{1}{p(\boldsymbol{Z}, \boldsymbol{Y})}\right] - \mathbb{E}\left[\log \frac{1}{p(\boldsymbol{\epsilon})}\right] \\
=& -\mathbb{E}\left[\log \frac{1}{p(\boldsymbol{\epsilon})}\right] \\
=& -H(\boldsymbol{\epsilon})
\end{aligned}
\tag{78}
$$

The entropy change is smaller than 0, therefore, the salt-and-pepper is a pure detrimental noise to the learning task.

### D.2 Add Salt-and-pepper Noise to Raw Images

The task entropy with salt-and-pepper noise is rewritten as:

$$
H(\mathcal{T}; \boldsymbol{X}\boldsymbol{\epsilon}) = -\sum_{\boldsymbol{Y}\in\mathcal{Y}} p(\boldsymbol{Y}|\boldsymbol{X}\boldsymbol{\epsilon}) \log p(\boldsymbol{Y}|\boldsymbol{X}\boldsymbol{\epsilon})
\tag{79}
$$

Since $\boldsymbol{\epsilon}$ is independent of $\boldsymbol{X}$ and $\boldsymbol{Y}$, the above equation can be expanded as:

$$
\begin{aligned}
H(\mathcal{T}; \boldsymbol{X}\boldsymbol{\epsilon}) =& -\sum_{\boldsymbol{Y}\in\mathcal{Y}} \frac{p(\boldsymbol{Y}, \boldsymbol{X}\boldsymbol{\epsilon})}{p(\boldsymbol{X})p(\boldsymbol{\epsilon})} \log \frac{p(\boldsymbol{Y}, \boldsymbol{X}\boldsymbol{\epsilon})}{p(\boldsymbol{X})p(\boldsymbol{\epsilon})} \\
=& -\sum_{\boldsymbol{Y}\in\mathcal{Y}} \frac{p(\boldsymbol{Y}, \boldsymbol{X})p(\boldsymbol{\epsilon})}{p(\boldsymbol{X})p(\boldsymbol{\epsilon})} \log \frac{p(\boldsymbol{Y}, \boldsymbol{X})p(\boldsymbol{\epsilon})}{p(\boldsymbol{X})p(\boldsymbol{\epsilon})} \\
=& -\sum_{\boldsymbol{Y}\in\mathcal{Y}} p(\boldsymbol{Y}|\boldsymbol{X}) \log p(\boldsymbol{Y}|\boldsymbol{X})
\end{aligned}
\tag{80}
$$

where

$$
\begin{aligned}
p(\boldsymbol{X}\boldsymbol{\epsilon}, \boldsymbol{Y}) =& p(\boldsymbol{X}\boldsymbol{\epsilon}|\boldsymbol{Y})p(\boldsymbol{Y}) \\
=& p(\boldsymbol{X}|\boldsymbol{Y})p(\boldsymbol{\epsilon}|\boldsymbol{Y})p(\boldsymbol{Y}) \\
=& p(\boldsymbol{X}|\boldsymbol{Y})p(\boldsymbol{\epsilon})p(\boldsymbol{Y}) \\
=& p(\boldsymbol{X}, \boldsymbol{Y})p(\boldsymbol{\epsilon})
\end{aligned}
\tag{81}
$$

Therefore, the entropy change with salt-and-pepper noise is:

$$
\triangle S(\mathcal{T}, Q\boldsymbol{X}) = H(\mathcal{T}; \boldsymbol{X}) - H(\mathcal{T}; \boldsymbol{X}\boldsymbol{\epsilon}) = 0
\tag{82}
$$

Salt-and-pepper noise can not help reduce the complexity of the task, and therefore, it is considered a type of pure detrimental noise.

From the discussion in this section, we can draw conclusions that **Linear Transform Noise** can be positive under certain conditions, while **Gaussian Noise** and **Salt-and-pepper Noise** are harmful noise. From the above analysis, the conditions that satisfy positive noise are forming a design space. Exploring the positive noise space is an important topic for future work.

Table 8: Details of ResNet Models. The columns "18-layer", "34-layer", "50-layer", and "101-layer" show the specifications of ResNet-18, ResNet-34, ResNet-50, and ResNet-101, separately.

| Layer name | Output size | 18-layer | 34-layer | 50-layer | 101-layer |
|---|---|---|---|---|---|
| conv1 | $112 \times 112$ | \multicolumn{4}{c}{$7 \times 7$, 64, stride 2} | | | |
| conv2_x | $56 \times 56$ | \multicolumn{4}{c}{$3 \times 3$, max pool, stride 2} | | | |
| conv2_x | $56 \times 56$ | $\begin{bmatrix} 3 \times 3 & 64 \\ 3 \times 3 & 64 \end{bmatrix} \times 2$ | $\begin{bmatrix} 3 \times 3 & 64 \\ 3 \times 3 & 64 \end{bmatrix} \times 3$ | $\begin{bmatrix} 1 \times 1 & 64 \\ 3 \times 3 & 64 \\ 1 \times 1 & 256 \end{bmatrix} \times 3$ | $\begin{bmatrix} 1 \times 1 & 64 \\ 3 \times 3 & 64 \\ 1 \times 1 & 256 \end{bmatrix} \times 3$ |
| conv3_x | $28 \times 28$ | $\begin{bmatrix} 3 \times 3 & 128 \\ 3 \times 3 & 128 \end{bmatrix} \times 2$ | $\begin{bmatrix} 3 \times 3 & 128 \\ 3 \times 3 & 128 \end{bmatrix} \times 4$ | $\begin{bmatrix} 1 \times 1 & 128 \\ 3 \times 3 & 128 \\ 1 \times 1 & 512 \end{bmatrix} \times 4$ | $\begin{bmatrix} 1 \times 1 & 128 \\ 3 \times 3 & 128 \\ 1 \times 1 & 512 \end{bmatrix} \times 4$ |
| conv4_x | $14 \times 14$ | $\begin{bmatrix} 3 \times 3 & 256 \\ 3 \times 3 & 256 \end{bmatrix} \times 2$ | $\begin{bmatrix} 3 \times 3 & 256 \\ 3 \times 3 & 256 \end{bmatrix} \times 6$ | $\begin{bmatrix} 1 \times 1 & 256 \\ 3 \times 3 & 256 \\ 1 \times 1 & 1024 \end{bmatrix} \times 6$ | $\begin{bmatrix} 1 \times 1 & 256 \\ 3 \times 3 & 256 \\ 1 \times 1 & 1024 \end{bmatrix} \times 23$ |
| conv5_x | $7 \times 7$ | $\begin{bmatrix} 3 \times 3 & 512 \\ 3 \times 3 & 512 \end{bmatrix} \times 2$ | $\begin{bmatrix} 3 \times 3 & 512 \\ 3 \times 3 & 512 \end{bmatrix} \times 3$ | $\begin{bmatrix} 1 \times 1 & 512 \\ 3 \times 3 & 512 \\ 1 \times 1 & 2048 \end{bmatrix} \times 3$ | $\begin{bmatrix} 1 \times 1 & 512 \\ 3 \times 3 & 512 \\ 1 \times 1 & 2048 \end{bmatrix} \times 3$ |
| | $1 \times 1$ | \multicolumn{4}{c}{average pool, 1000-d fc, softmax} | | | |
| Params | | 11M | 22M | 26M | 45M |

Table 9: Details of ViT Models. Each row shows the specifications of a kind of ViT model. ViT-T, ViT-S, ViT-B, and ViT-L represent ViT Tiny, ViT Small, ViT Base, and ViT Large, separately.

| ViT Model | Layers | Hidden size | MLP size | Heads | Params |
|---|---|---|---|---|---|
| ViT-T | 12 | 192 | 768 | 3 | 5.7M |
| ViT-S | 12 | 384 | 1536 | 6 | 22M |
| ViT-B | 12 | 768 | 3072 | 12 | 86M |
| ViT-L | 24 | 1024 | 4096 | 16 | 307M |

# E  Experimental Setting

We introduce the implementation details in this part. The noise was added to both the training stage and the inference stage. Model details are shown in Table 8 and 9. The image resolution is $224 \times 224$ for all the experiments. Pre-trained models on ImageNet-21K are used as the backbone. We train all ResNet and ViT-based models using AdamW optimizer Loshchilov & Hutter (2017). We set the learning rate of each parameter group using a cosine annealing schedule with a minimum of $1e - 7$. The data augmentation for training only includes the random resized crop and normalization.

**CNN(ResNet) Setting** The training epoch is set to 100. We initialized the learning rate as 0 and linearly increase it to 0.001 after 10 warmup steps. All the experiments of CNNs are trained on a single Tesla V100 GPU with 32 GB. The batch size for ResNet18, ResNet34, ResNet50, and ResNet101 are 1024, 512, 256, and 128, respectively.

**ViT and Variants Setting** All the experiments of ViT and its variants are trained on a single machine with 8 Tesla V100 GPUs. For vanilla ViTs, including ViT-T, ViT-S, ViT-B, and ViT-L, the training epoch is set to 50 and the input patch size is $16 \times 16$. We initialized the learning rate as 0 and linearly increase it to 0.0001 after 10 warmup steps. We then decrease it by the cosine decay strategy. For experiments on the variants of ViT, the training epoch is set to 100 and the learning rate is set to 0.0005 with 10 warmup steps.

# F  More Experiment Results

We present additional experimental results demonstrating the effects of injecting positive noise into various ViT series variants, including SwinTransformer, DeiT, ConViT, and BeiT, using the Tiny ImageNet Le & Yang (2015b) dataset. The positive noise intensity is set to 0.3, and the noise is applied to the final layer.

Additionally, we report the performance of NoisyNN across multiple datasets, including Tiny ImageNet Le & Yang (2015b), ImageNet-A Hendrycks et al. (2021), ImageNet-C Hendrycks & Dietterich (2019), CI-FAR Krizhevsky (2009), and INbreast Moreira et al. (2012). Furthermore, we compare the proposed method with common data augmentation techniques, alternative noise types, and Manifold MixUp Verma et al. (2019). Lastly, we showcase the application of our method to tasks such as domain generalization and text classification, highlighting its versatility and effectiveness.

## F.1   Inject Positive Noise to Variants of ViT

As demonstrated in the paper, the positive noise can be injected into the ViT family. Therefore, in this section, we explore the influence of positive noise on the variants of the ViT. The positive noise used here is identical to that in the paper. For this, we comprehensively compare noise injection to ConViT d'Ascoli et al. (2021), BeiT Bao et al. (2021), DeiT Touvron et al. (2021), and Swin Transformer Liu et al. (2021), and comparisons results are reported in Tabel 10. As expected, these variants of ViTs get benefit from the positive noise. The additional four ViT variants are at the base scale, whose parameters are listed in the table's last row. For a fair comparison, we use identical experimental settings for each kind of experiment. For example, we use the identical setting for vanilla ConViT, ConViT with different kinds of noise. From the experimental results, we can observe that the different variants of ViT benefit from positive noise and significantly improve prediction accuracy. The results on different scale datasets and variants of the ViT family demonstrate that positive noise can universally improve the model performance by a wide margin.

Table 10: Variants of ViT with different kinds of noise on TinyImageNet. Vanilla means the vanilla model without noise. Accuracy is shown in percentage. Gaussian noise used here is subjected to standard normal distribution. Linear transform noise used in this table is designed to be positive noise. The difference is shown in the bracket.

| Model | DeiT | SwinTransformer | BeiT | ConViT |
|---|---|---|---|---|
| Vanilla | 85.02 (+0.00) | 90.84 (+0.00) | 88.64 (+0.00) | 90.69 (+0.00) |
| + Gaussian Noise | 84.70 (-0.32) | 90.34 (-0.50) | 88.40 (-0.24) | 90.40 (-0.29) |
| + Linear Transform Noise | **86.50 (+1.48)** | **95.68 (+4.84)** | **91.78 (+3.14)** | **93.07 (+2.38)** |
| + Salt-and-pepper Noise | 84.03 (-1.01) | 87.12 (-3.72) | 42.18 (-46.46) | 89.93 (-0.76) |
| Params. | 86M | 87M | 86M | 86M |

## F.2   Positive Noise on TinyImageNet

We also implement experiments of ResNet and ViT on the smaller dataset TinyImageNet, and the results are shown in Table 11 and 12. As shown in the tables, positive noise also benefits the deep models on the small dataset. From the experiment results of CNN and ViT family on ImageNet and TinyImageNet, we can find that the positive noise has better effects on larger datasets than smaller ones. This makes sense because as shown in the section on optimal quality matrix, the upper boundary of the entropy change is determined by the size, i.e., the number of data samples, of the dataset, smaller datasets have less number of data samples, which means the upper boundary of the small datasets is lower than the large datasets. Therefore, the positive noise of linear transform noise has better influences on large than small datasets.

## F.3   Positive Noise on ImageNet-A

Table 13 shows additional results on ImageNet-A. We further tested the positive linear transformation noise on ImageNet-A, which exhibits a significant domain shift compared to the validation set of ImageNet-1k. The results demonstrate the robustness of our method to domain shift. We also calculate the confusion matrices of our method and ViT-B on ImageNet-A, which are presented in Fig. 3a and 3b, respectively.

Table 11: ResNet with different kinds of noise on TinyImageNet. Vanilla means the vanilla model without noise. Accuracy is shown in percentage. Gaussian noise used here is subjected to standard normal distribution. Linear transform noise used in this table is designed to be positive noise. The difference is shown in the bracket.

| Model | ResNet-18 | ResNet-34 | ResNet-50 | ResNet-101 |
|---|---|---|---|---|
| Vanilla | 64.01 (+0.00) | 67.04 (+0.00) | 69.47 (+0.00) | 70.66 (+0.00) |
| + Gaussian Noise | 63.23 (-0.78) | 65.71 (-1.33) | 68.17 (-1.30) | 69.13 (-1.53) |
| + Linear Transform Noise | **73.32 (+9.31)** | **76.70 (+9.66)** | **76.88 (+7.41)** | **77.30 (+6.64)** |
| + Salt-and-pepper Noise | 55.97 (-8.04) | 63.52 (-3.52) | 49.42 (-20.25) | 53.88 (-16.78) |

Table 12: ViT with different kinds of noise on TinyImageNet. Vanilla means the vanilla model without injecting noise. Accuracy is shown in percentage. Gaussian noise used here is subjected to standard normal distribution. Linear transform noise used in this table is designed to be positive noise. The difference is shown in the bracket. Note **ViT-L is overfitting on TinyImageNet** Dosovitskiy et al. (2020) Steiner et al. (2021).

| Model | ViT-T | ViT-S | ViT-B | ViT-L |
|---|---|---|---|---|
| Vanilla | 81.75 (+0.00) | 86.78 (+0.00) | 90.48 (+0.00) | 93.32 (+0.00) |
| + Gaussian Noise | 80.95 (-0.80) | 85.66 (-1.12) | 89.61 (-0.87) | 92.31 (-1.01) |
| + Linear Transform Noise | **82.50 (+0.75)** | **91.62 (+4.84)** | **94.92 (+4.44)** | **93.63 (+0.31)** |
| + Salt-and-pepper Noise | 79.34 (-2.41) | 84.66 (-2.12) | 87.45 (-3.03) | 83.48 (-9.84) |

Table 13: Top 1 accuracy on ImageNet-A with positive linear transform noise.

| Model | Top1 Acc. | Params. | Image Res. | Pretrained Dataset |
|---|---|---|---|---|
| ViT-B | 27.4 | 86M | $224 \times 224$ | ImageNet 21k |
| NoisyNN (ViT-B based) | 34.1 | 86M | $224 \times 224$ | ImageNet 21k |
| NoisyNN (ViT-B based) | 38.3 | 348M | $384 \times 384$ | ImageNet 21k |

### F.4 Positive Noise on ImageNet-C

Table 14 shows additional results on ImageNet-C. ImageNet-C exhibits various forms of domain shift in comparison to the validation set of ImageNet-1k. The results further demonstrate the robustness of our method to such domain shifts.

### F.5 CIFAR and INbreast Results

Results on CIFAR-10, CIFAR-100, and INbreast are shown in Table 15. Showing the effectiveness of NoisyNN beyond ImageNet-based datasets.

### F.6 Comparison and Combination with Common Data Augmentation Techniques

We compare our method with common data augmentation methods, and the results are presented in Table 17. Additionally, we combine our method with data augmentations, and the corresponding results are shown in Table 16.

### F.7 Comparison with Other Noises

Below in Table 18 we compare NoisyNN to other commonly seen noises including White Noise, Uniform Noise and Dropout (Srivastava et al., 2014) on TinyImageNet.

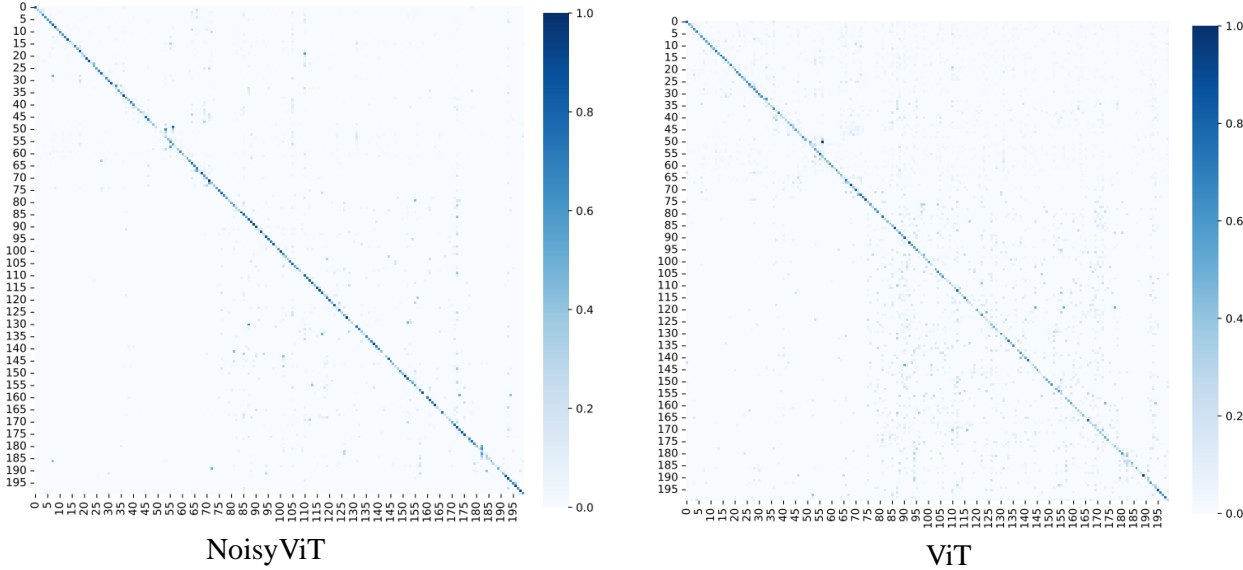

(a) Confusion Matrix of NoisyNN on ImageNet-A.     (b) Confusion Matrix of ViT on ImageNet-A.

Figure 3: Comparison of Confusion Matrices: NoisyNN (ViT-based) and ViT on ImageNet-A.

Table 14: Top 1 accuracy on ImageNet-C with positive linear transform noise.

| Model | Top1 Acc. | Params. | Image Res. | Pretrained Dataset |
|---|---|---|---|---|
| ViT-B | 53.4 | 86M | $224 \times 224$ | ImageNet 21k |
| NoisyNN (ViT-B based) | 58.1 | 86M | $224 \times 224$ | ImageNet 21k |
| NoisyNN (ViT-B based) | 60.5 | 348M | $384 \times 384$ | ImageNet 21k |

Table 15: Comparing ViT-B with NoisyNN on CIFAR-10, CIFAR-100 and INbreast.

| Model | CIFAR-100 | CIFAR-10 | INbreast |
|---|---|---|---|
| ViT-B | 91.5±0.1 | 98.6±0.1 | 90.6±0.2 |
| NoisyNN (ViT-B based) | **93.7±0.1** | **99.4±0.1** | **93.5±0.1** |

Table 16: Combining NoisyNN with Data Augmentation.

| Method | ImageNet |
|---|---|
| NoisyNN (No DA) | **89.9±0.5** |
| NoisyNN + RandomResizedCrop | 89.1±0.5 |
| NoisyNN + RandomHorizontalFlip+RandomResizedCrop | 89.2±0.6 |
| NoisyNN + RandomResizedCrop+RandAugment | 89.4±0.5 |

Table 17: Comparing NoisyNN with Data Augmentation.

| Method | ImageNet |
|---|---|
| ViT-B | 84.3 |
| ViT-B+RandomFlip+Gaussian Blur | 84.2 |
| ViT-B+RandAugment | 85.1 |
| ViT-B+Linear Transformation Noise (NoisyNN) | **89.9** |

Table 18: Comparison of NoisyNN with other noises on TinyImageNet.

|  | ResNet18 | ResNet34 | ResNet50 |
|---|---|---|---|
| Vanilla | 64.01 | 67.04 | 69.47 |
| White Noise | 64.05 | 65.97 | 68.87 |
| Uniform Noise | 64.05 | 66.01 | 69.01 |
| Gaussian Noise | 63.23 | 64.71 | 68.17 |
| Salt-and-pepper | 55.97 | 63.52 | 49.42 |
| Dropout | 63.96 | 67.01 | 69.40 |
| NoisyNN (ours) | **73.32** | **76.70** | **76.88** |

## F.8 Positive Noise for Domain Generalization

Domain Generalization (DG) methods try to learn a robust model by training on multiple source domains Volpi et al. (2018); Seo et al. (2020); Carlucci et al. (2019); Huang et al. (2020), while DG methods cannot access the target domains during the training stage. To verify our method in the application of DG tasks, we further conduct experiments on VLCS and PACS, two commonly used datasets in the field of DG. The results are reported in Table 19. As shown in the table, compared to competitive methods, our proposed method achieves state-of-the-art (SOTA) results on the PACS and VLCS datasets.

Table 19: Comparison with other methods in domain generalization tasks.

| Method | PACS | VLCS |
|---|---|---|
| ViT Dosovitskiy et al. (2020) (ICLR'21) | 85.0 | 76.9 |
| SDViT (Sultana et al., 2022) (ACCV'22) | 88.9 | 81.9 |
| ALOFT (Guo et al., 2023) (CVPR'23) | 91.6 | 81.3 |
| NoisyViT | **93.1** | **84.4** |

## F.9 Positive Noise for Text Classification

Text classification involves categorizing text into predefined classes or labels (Kowsari et al., 2019). It is widely used in various applications such as spam detection, sentiment analysis, topic labeling, and document categorization. To check whether our method can be applied to a different data modality but within the same problem of classification, we conduct experiments on two popular text classification datasets with widely used models. The results are shown in Table 20. Equipped with our method, TextCNN and TextRNN show a significant improvement in performance.

Table 20: Comparison with other methods in text classification tasks.

| Method | THUNews | AGNews |
|---|---|---|
| TextCNN (Kim, 2014) (EMNLP'14) | 90.8 | 89.2 |
| NoisyTextCNN | **93.4** | **89.3** |
| TextRNN (Liu et al., 2016) (IJCAI'16) | 90.7 | 87.7 |
| NoisyTextRNN | **95.5** | **88.1** |

## F.10 Positive Noise for Domain Adaptation

Unsupervised domain adaptation (UDA) aims to learn transferable knowledge across the source and target domains with different distributions Pan & Yang (2009) Wei et al. (2018). There are mainly two kinds of deep neural networks for UDA, which are CNN-based and Transformer-based methods Sun et al. (2022) Yang et al. (2023a). Various techniques for UDA are adopted on these backbone architectures. For example, the discrepancy techniques measure the distribution divergence between source and target domains Long et al. (2018) Sun & Saenko (2016). Adversarial adaptation discriminates domain-invariant and domain-specific

Table 21: Comparison with SOTA methods (ResNet-50 He et al. (2016), MinEnt Grandvalet & Bengio (2004), SAFN Xu et al. (2019), CDAN+E Long et al. (2018), DCAN Li et al. (2020), BNM Cui et al. (2020), SHOT Liang et al. (2020), ATDOC-NA Liang et al. (2021), ViT-B Dosovitskiy et al. (2020), TVT-B Yang et al. (2023a), CDTrans-B Xu et al. (2022), SSRT-B Sun et al. (2022)) on **Office-Home**. The best performance is marked in red.

| Method | Ar2Cl | Ar2Pr | Ar2Re | Cl2Ar | Cl2Pr | Cl2Re | Pr2Ar | Pr2Cl | Pr2Re | Re2Ar | Re2Cl | Re2Pr | Avg. |
|---|---|---|---|---|---|---|---|---|---|---|---|---|---|
| ResNet-50 | 44.9 | 66.3 | 74.3 | 51.8 | 61.9 | 63.6 | 52.4 | 39.1 | 71.2 | 63.8 | 45.9 | 77.2 | 59.4 |
| MinEnt | 51.0 | 71.9 | 77.1 | 61.2 | 69.1 | 70.1 | 59.3 | 48.7 | 77.0 | 70.4 | 53.0 | 81.0 | 65.8 |
| SAFN | 52.0 | 71.7 | 76.3 | 64.2 | 69.9 | 71.9 | 63.7 | 51.4 | 77.1 | 70.9 | 57.1 | 81.5 | 67.3 |
| CDAN+E | 54.6 | 74.1 | 78.1 | 63.0 | 72.2 | 74.1 | 61.6 | 52.3 | 79.1 | 72.3 | 57.3 | 82.8 | 68.5 |
| DCAN | 54.5 | 75.7 | 81.2 | 67.4 | 74.0 | 76.3 | 67.4 | 52.7 | 80.6 | 74.1 | 59.1 | 83.5 | 70.5 |
| BNM | 56.7 | 77.5 | 81.0 | 67.3 | 76.3 | 77.1 | 65.3 | 55.1 | 82.0 | 73.6 | 57.0 | 84.3 | 71.1 |
| SHOT | 57.1 | 78.1 | 81.5 | 68.0 | 78.2 | 78.1 | 67.4 | 54.9 | 82.2 | 73.3 | 58.8 | 84.3 | 71.8 |
| ATDOC-NA | 58.3 | 78.8 | 82.3 | 69.4 | 78.2 | 78.2 | 67.1 | 56.0 | 82.7 | 72.0 | 58.2 | 85.5 | 72.2 |
| ViT-B | 54.7 | 83.0 | 87.2 | 77.3 | 83.4 | 85.6 | 74.4 | 50.9 | 87.2 | 79.6 | 54.8 | 88.8 | 75.5 |
| TVT-B | 74.9 | 86.8 | 89.5 | 82.8 | 88.0 | 88.3 | 79.8 | 71.9 | 90.1 | 85.5 | 74.6 | 90.6 | 83.6 |
| CDTrans-B | 68.8 | 85.0 | 86.9 | 81.5 | 87.1 | 87.3 | 79.6 | 63.3 | 88.2 | 82.0 | 66.0 | 90.6 | 80.5 |
| SSRT-B | 75.2 | 89.0 | 91.1 | 85.1 | 88.3 | 90.0 | 85.0 | 74.2 | 91.3 | 85.7 | 78.6 | 91.8 | 85.4 |
| NoisyTVT-B | **78.3** | **90.6** | **91.9** | **87.8** | **92.1** | **91.9** | **85.8** | **78.7** | **93.0** | **88.6** | **80.6** | **93.5** | **87.7** |

representations by playing an adversarial game between the feature extractor and a domain discriminator Ganin & Lempitsky (2015).

Recently, transformer-based methods achieved SOTA results on UDA, therefore, we evaluate the ViT-B with the positive noise on widely used UDA benchmarks. Here the positive noise is the linear transform noise identical to that used in the classification task. The positive noise is injected into the last layer of the model, the same as the classification task. The datasets include **Office Home** Venkateswara et al. (2017) and **VisDA2017** Peng et al. (2017). **Office-Home** Venkateswara et al. (2017) has 15,500 images of 65 classes from four domains: Artistic (Ar), Clip Art (Cl), Product (Pr), and Real-world (Rw) images. **VisDA2017** is a Synthetic-to-Real object recognition dataset, with more than 0.2 million images in 12 classes. We use the ViT-B with a $16 \times 16$ patch size, pre-trained on ImageNet. We use minibatch Stochastic Gradient Descent (SGD) optimizer Ruder (2016) with a momentum of 0.9 as the optimizer. The batch size is set to 32. We initialized the learning rate as 0 and linearly warm up to 0.05 after 500 training steps. The results are shown in Table 21 and 22. The methods above the black line are based on CNN architecture, while those under the black line are developed from the Transformer architecture. The NoisyTVT-B, i.e., TVT-B with positive noise, achieves better performance than existing works. These results show that positive noise can improve model generality and, therefore, benefit deep models in domain adaptation tasks.

Table 22: Comparison with SOTA methods on **Visda2017**. The best performance is marked in red.

| Method | plane | bcycl | bus | car | horse | knife | mcycl | person | plant | sktbrd | train | truck | Avg. |
|---|---|---|---|---|---|---|---|---|---|---|---|---|---|
| ResNet-50He et al. (2016) | 55.1 | 53.3 | 61.9 | 59.1 | 80.6 | 17.9 | 79.7 | 31.2 | 81.0 | 26.5 | 73.5 | 8.5 | 52.4 |
| DANNGanin & Lempitsky (2015) | 81.9 | 77.7 | 82.8 | 44.3 | 81.2 | 29.5 | 65.1 | 28.6 | 51.9 | 54.6 | 82.8 | 7.8 | 57.4 |
| MinEntGrandvalet & Bengio (2004) | 80.3 | 75.5 | 75.8 | 48.3 | 77.9 | 27.3 | 69.7 | 40.2 | 46.5 | 46.6 | 79.3 | 16.0 | 57.0 |
| SAFNXu et al. (2019) | 93.6 | 61.3 | 84.1 | 70.6 | 94.1 | 79.0 | 91.8 | 79.6 | 89.9 | 55.6 | 89.0 | 24.4 | 76.1 |
| CDAN+ELong et al. (2018) | 85.2 | 66.9 | 83.0 | 50.8 | 84.2 | 74.9 | 88.1 | 74.5 | 83.4 | 76.0 | 81.9 | 38.0 | 73.9 |
| BNM Cui et al. (2020) | 89.6 | 61.5 | 76.9 | 55.0 | 89.3 | 69.1 | 81.3 | 65.5 | 90.0 | 47.3 | 89.1 | 30.1 | 70.4 |
| CGDMDu et al. (2021) | 93.7 | 82.7 | 73.2 | 68.4 | 92.9 | 94.5 | 88.7 | 82.1 | 93.4 | 82.5 | 86.8 | 49.2 | 82.3 |
| SHOTLiang et al. (2020) | 94.3 | 88.5 | 80.1 | 57.3 | 93.1 | 93.1 | 80.7 | 80.3 | 91.5 | 89.1 | 86.3 | 58.2 | 82.9 |
| ViT-BDosovitskiy et al. (2020) | 97.7 | 48.1 | 86.6 | 61.6 | 78.1 | 63.4 | 94.7 | 10.3 | 87.7 | 47.7 | 94.4 | 35.5 | 67.1 |
| TVT-BYang et al. (2023a) | 92.9 | 85.6 | 77.5 | 60.5 | 93.6 | 98.2 | 89.4 | 76.4 | 93.6 | 92.0 | 91.7 | 55.7 | 83.9 |
| CDTrans-BXu et al. (2022) | 97.1 | 90.5 | 82.4 | 77.5 | 96.6 | 96.1 | 93.6 | **88.6** | **97.9** | 86.9 | 90.3 | 62.8 | 88.4 |
| SSRT-B Sun et al. (2022) | **98.9** | 87.6 | **89.1** | **84.8** | 98.3 | **98.7** | **96.3** | 81.1 | 94.9 | 97.9 | 94.5 | 43.1 | 88.8 |
| NoisyTVT-B | 98.8 | **95.5** | 84.8 | 73.7 | **98.5** | 97.2 | 95.1 | 76.5 | 95.9 | **98.4** | **98.3** | **67.2** | **90.0** |

