# OpenReview forum: "NoisyNN: Exploring the Impact of Information Entropy Change in Learning Systems"
_TMLR — Rejected by TMLR_

### Review · Reviewer_yo8p · 2025-01-24

**Summary Of Contributions:**

The paper studies noise injection in image classification. That is, can we improve performance by injecting noise on the layer features, ie. the layer’s (NCHW) embeddings. They show that minimizing the predictive entropy of the classifier (ie. uncertainty), one can get massive accuracy boost.

The authors assume that task complexity can be measured as the entropy of the predictive distribution. They then show that adding salt-and-pepper or Gaussian noise to layer signal increases the entropy, which should decrease performance. In contrast, linearly transforming the layer features can reduce the entropy, and the largest reduction is achieved by a graph Laplacian type diffusion matrix that spreads the latent features between dimensions and between datapoints. This effectively smoothens the layer features. The authors also propose a circular transform, which only spreads along one axis.

**Audience:**

Yes

**Broader Impact Concerns:**

No issues

**Claims And Evidence:**

Yes

**Requested Changes:**

While the authors present the method as noise injection, there does not seem to be any noise [eq 17 is not used anywhere], and it is more akin to a combination of layernorm and batchnorm. That is, the signal is being globally blurred over the layer’s (NCHW) tensor. It is hard to see why this would be a good idea, or why it reduces entropy. I might be wrong here: the paper is vague so I’m not sure if this actually happens.

Furthermore, it’s not clear if reducing entropy is good. Predictive entropy also manages the predictive uncertainty, and by decreasing it we are overfitting the model more and more. While this can obviously be helpful in terms of training error and training faster wrt the training dataset, surely it should just worsen test error? The authors should discuss the train/test errors and generalisation, and also should show how the training loss evolves during training.

I find the paper presentation to be sloppy in terms of math: below is a list of issues. The method itself is unclear: the author’s need to add an architecture diagram or algorithm box describing how their noise actually changes the model/network/pipeline. The authors say that they add this to the last layer. Ok, but what does that mean? Before or after the last layer, and what even is the last layer (is it the softmaxes, logits, or something else)? It’s also unclear when and how much noise is added (if any): do we just apply Q once or multiple times?

How does the noise injection interplay with the loss?

The code needs to be included.

The authors show huge improvements of 10..15 percentage points in ImageNet benchmark, and achieve top-1 accuracy of up to 95.65%.  https://paperswithcode.com/sota/image-classification-on-imagenet shows current world record at 91%. If the results are valid, this paper then surpasses this by a clear margin.

This obviously warrants healthy scepticism, and the authors do not discuss the significance of the results. Is this really a new state of the art? Are the results comparable? The authors need to discuss this further, and convince that no information leakage is happening, and that test errors are being properly computed.

The main problem with the result is their black-boxness. The results are not opened up, nor any insight drawn on how we improve or why. I’m quite confused what the method even does. This needs clarifications.


- Throughout the paper random variables and their densities are not clearly defined, and there is no clear definition whether we work with random variables or random vectors or random matrices. It’s not clear if the inputs or outputs are vectorial.
- The notation is inconsistent: we use “x” for random variable, but also “X”. The notation needs to be clear in terms of scalars/vectors/matrices.
- The “task” is undefined, and measuring an entropy of task has no clear meaning. This only makes sense if task is defined as the predictive density p(y|x).
- Even if we think of tasks as conditional densities, I wonder if entropy is a sensible measure of task complexity. In classification tasks we often have a hard boundary where the true underlying system can classify things perfectly. In such case the entropy is zero. Yet, this tells us nothing whether the tasks is simple or complex (ie. whether the decision boundary is simple or complex). The entropy would then only measure effectively the label variability. For instance, one could have similar amount of label variability in MNIST as in ImageNet (ie. ambivalent corner cases), even though they have wildly different complexities as tasks. Given all of this, I would then disagree that entropy is a useful measure of task difficult, as stated before eq 4. I’m interested to hear author’s response.
- In eq 4 we go further and define H(T|eps). There is no definition of this, and I’m pretty confused what this means. Is this perhaps H[p(T|eps)] or H[p(T)|eps]? The paper needs be more rigorous in terms of random variables and their densities.
- I wonder what it conceptually means if the task entropy goes down through entropy. Does the task become easier, or do the label distributions narrow? If the label distribution narrows beyond the true distribution, this just overfits.
- The notation should also make it clear when the densities are model predictive and when true predictive.
- The nature of bf{X} and bf{Y} are unclear: are these matrices?
- In eq 6 LHS has T and X, but RHS has (T,X,Y). This is incorrect: Y can’t just appear on left, since it will be undefined.
- Eq 6 seems a bit strange. Is this a definition or derivation? The H(Y|X)-H(X) is just mutual information H(X,Y), which doesn’t have the task anymore. Where did the task go on RHS? This needs proper derivation.
- Eq 6 has no citation, nor a proof, nor a derivation. Can you provide any?
- I don’t understand the jump from eq 6 to 7. In eq 6 we define the conditional entropy as two terms, but in eq 7 it’s just one term. Where did the H(X) disappear in eq 7?
- Also, according to eq 3 the conditional entropy is \int p(y,x) log p(y|x), while in eq 7 it is \int p(y|x) log p(y|x). That is, it seems that eq 7 is an entropy of a conditional distribution; while eq 3 is conditional entropy. These are two different things, and it seems that the notation is confusing these.
- The “task” in eq 8 gets even more confusing. Now the task is going from some latents to labels. These latents are not part of the original task, so this is then some new variant of the original task. For instance, in MNIST the task is to classify digit images as humans do [that is, humans are the ground truth]. Now we have changed the task to classify some arbitrary vectors of latent numbers instead, which no human could do. The meaning of “task” should be clarified. Also, one needs to clarify why the “T” disappears in eq 8 RHS.
- Eq 9 uses different notation than eq 7, even though they should be the same. Can we compare these two? It seems that in eq 9 we have the entropies H(Z), but in eq 7 the corresponding entropy H(X) is not present (or is it?). It would be clearer to harmonise  notation.
- After eq 10 we assume that Y and Z are Gaussians. This is a drastic assumption: surely they are not. Can you comment on this?
- Y and Z are matrices. What does it then mean that they are Gaussian? Do the matrices follow some matrix-variate gaussian, or do the columns or rows follow vector-valued Gaussians? Be precise.
- I’m not sure how you move in eq 11 from second line to third, or to fourth.
- What does variance of data sample mean? Data samples are not random variables: they have no variance. The distribution they are sampled from has variance.
- The nature of covariances and \Sigma needs to be made precise. What does a covariance between two matrices mean? I think this is actually a “sample cross-covariance”. Furthermore, assuming that Z_i and Y_i are vectors, the cov(Z_i,Y_i) will become a cross-covariance matrix. How do you handle this? Please define the cov(Zi,Yi) precisely.
- What does neighbor Z_{i+1} mean? What are the shapes of Z and Q? The circulation goes either within each latent vector, or between latent vectors of different datapoints. The latter sounds strange, so I assume it’s inside a latent vector. Does the Q stuff have any noise or randomness? It does not seem to.
- How is the noise actually injected? Does it happen once per minibatch? Is it only on one layer? Please include an algorithm box.

**Strengths And Weaknesses:**

S: The experiments are comprehensive, and show fantastic performance
S: The idea is novel and fresh
W: The math presentation has some issues
W: The paper is quite black-box, and offers little insight into how the method works or why it improves performance

---

### Review · Reviewer_gFAz · 2025-02-14

**Summary Of Contributions:**

This paper has two contributions: first it analyses mathematically the effect of input and embedding noise on the conditional entropy of the label distribution. Arguing that this reflects the complexity of a task, it then suggests noise distributions that would decrease, rather than increase, this complexity. This so-called positive noise is then applied on ResNet and ViT models trained on ImageNet, and other datasets, showing superior performance.

**Audience:**

Yes

**Broader Impact Concerns:**

No concerns.

**Claims And Evidence:**

Yes

**Requested Changes:**

Strengths:
- The idea of tying label entropy to difficulty is interesting, and showing that it is a controllable quantity is valuable
    - I especially appreciate the insights that tie non-0 conditional label entropy to past works
- Surprisingly strong empirical results

Weaknesses:
- I'm having a lot of trouble following the notation used in the paper, and have some concerns about the assumptions made in the derivations
- I also have concerns about how the paper is positioning itself with respect to past work, which I find dismisses (albeit perhaps unintentionally?) decades of past work
- Although the theoretical justification (which I have concerns about) is presumably able to explain that the intervention makes sense, there is still a gap between
   - complexity and accuracy: explaining why a lower task complexity means a better accuracy
   - theory and practice: empirically verifying that the proposed intervention does indeed change the entropy as suggested by theory.


This last gap between theory and practice is especially a concern for me. Let's say I wrote a paper about how L1 regularization on DNN parameters helps generalization. My hypothesis would be "more near-0-parameters is good", my mechanism would be L1. Therefore I would need to measure two, maybe three, things, (a) that there is a link between L1 strength and generalization (this paper has that), (b) that there is a link between L1 strength and number of "near-0-parameters", and (c\) that it is the same link between number of "near-0-parameters" and generalization (this paper does not have (b) or (c\)). (c\) normally follows from (a) and (b), but not always.

Additionally, this is a concern because, going back to position with respect to past work, there is a general consensus that there are perturbations we can add to neural networks that improve their performance. Such perturbations have a variety of rationales; it could be that the proposed Linear Tranform Noise benefits models for reasons unrelated to task complexity.

### Concerns about notation

I'm having trouble making sense of the definition of task complexity. There seems to be some very unconventional overload of $H$. $H(T;X)$ is defined in (6) as

$$H(T;X) := H(Y;X)-H(X)$$

at this point it's not clear to me what $H(Y;X)$ is. Looking at (7), $H(A; B)$ doesn't seem to be conditional entropy $H(A|B)$, but rather the entropy $H(A|B=b)$, or in our case $H(\mathbf{Y}|\mathcal{X}=\mathbf{X})$ (but where did $H(X)$ go?). Then later in (9) the notation once again suggests $H(T;X) := H(Y;X)-H(X)$

Appendix A adds to my confusion, as it defines
$$H(T; Z) := H(Y, Z) - H(Z)$$
i.e. using the joint entropy, but looking at say equation (33), the fourth step clearly substitutes $H(Y; Z)$ with $H(Y|Z) + H(Z)$, and looking at equation (29), $H(T;Z)$ is equated with $H(Y|Z=z)$

Again going back to (33), we see that $H(T;Z) = H(Y;Z) - H(Z)$ and $H(Y;Z) = H(Y|Z) + H(Z)$ and so then $H(T;Z) = H(Y|Z) + H(Z) - H(Z) = H(Y|Z) \neq H(Y|Z=z)$.

I would ask the authors to use a clearer notation, specifically clearly denote the use of conditional entropies vs "entropies given X=x". These are well defined and I don't see the need to introduce new notation.

### Concerns about assumptions

Trying to understand the details in appendix A & B I have some further questions:
- How is it valid to assume $Y$ follows a multivariate distribution, but then train a classifier?
- There seems to be an assumption that the embeddings $Z$ are independent across samples, but deep learning embeddings are obtained via learned non-linear transforms. This learning process _should_ induce some correlations between embeddings of different samples. I'm not sure how this is accounted for here.

### Text notes

After equation (19), "where $p$ is a probability generated by a random seed", what is the distribution of $p$? Uniform?

Unless I'm misunderstanding something, the authors suggest that the deeper the layer used to add noise is the better, but only go up to layer 4 for CNNs (Fig 2b). Is there a reason why?


### Concerns about positioning

The authors write that "Noise is conventionally viewed as a harmful perturbation", "Our work [..] reveals a surprising fact: the simple injection of noise into deep neural networks [..] can significantly enhance model performance." I really don't think this is an accurate portrayal of convention.

Dropout (2012) is an ubiquitous strategy in CNNs and Transformers; diffusion models are literally using noise to learn to generate data; image noise is a common data augmentation strategy that can boost the generalization performance of image models (earliest instance I can think of is LetNet-5, 1998). Manifold mixup is also a clear example of latent space noise, that the authors acknowledge.

I appreciate that the authors are proposing a novel theoretical angle on the question, but the picture painted here dismisses decades of work on the question. It's ok to acknowledge the existence of past methods positively and to phrase one's contribution as additive. There is no need to deform the reality of prior work to make one's work appear shocking and "revealing surprising facts". This is doubly concerning as the authors clearly acknowledge some of the past deep learning literature that uses noise in beneficial ways.

**Strengths And Weaknesses:**

Strengths:
- The idea of tying label entropy to difficulty is interesting, and showing that it is a controllable quantity is valuable
    - I especially appreciate the insights that tie non-0 conditional label entropy to past works
- Surprisingly strong empirical results

Weaknesses:
- I'm having a lot of trouble following the notation used in the paper, and have some concerns about the assumptions made in the derivations
- I also have concerns about how the paper is positioning itself with respect to past work, which I find dismisses (albeit perhaps unintentionally?) decades of past work
- Although the theoretical justification (which I have concerns about) is presumably able to explain that the intervention makes sense, there is still a gap between
   - complexity and accuracy: explaining why a lower task complexity means a better accuracy
   - theory and practice: empirically verifying that the proposed intervention does indeed change the entropy as suggested by theory.


This last gap between theory and practice is especially a concern for me. Let's say I wrote a paper about how L1 regularization on DNN parameters helps generalization. My hypothesis would be "more near-0-parameters is good", my mechanism would be L1. Therefore I would need to measure two, maybe three, things, (a) that there is a link between L1 strength and generalization (this paper has that), (b) that there is a link between L1 strength and number of "near-0-parameters", and (c\) that it is the same link between number of "near-0-parameters" and generalization (this paper does not have (b) or (c\)). (c\) normally follows from (a) and (b), but not always.

Additionally, this is a concern because, going back to position with respect to past work, there is a general consensus that there are perturbations we can add to neural networks that improve their performance. Such perturbations have a variety of rationales; it could be that the proposed Linear Tranform Noise benefits models for reasons unrelated to task complexity.

---

> ### Author Response · Authors · 2025-03-21
>
> Thank you for your insightful comments, which greatly improve the quality of our work. We appreciate the opportunity to address your concerns and provide clarifications.
>
> **Concern about the logical connection between theory and practice.**
>
> The positive noise paper [1] demonstrated that reducing task complexity by adding noise can improve model performance in classification tasks. The positive noise framework suggests that task complexity corresponds to the difficulty of a learning task, lower complexity leads to reduced difficulty, which in turn results in higher accuracy. Our work builds on this existing framework [1] but differs by adding noise to embeddings instead of raw images.
>
> While various perturbations can enhance neural network performance for different reasons, our paper specifically adopts the concept of task entropy from the positive noise framework to explain this phenomenon. Our empirical findings show that adding linear transformation noise benefits the model, and the positive noise framework explains this phenomenon well. More importantly, by leveraging task entropy, we can predict the optimal form of linear transformation noise in Eq. 20, and this prediction is validated through our experiments in Table 5.
>
> **Concerns about notation**
>
> In our work, we analyze the uncertainty in predicting labels Y given a fixed dataset or set of features, i.e., $H(Y|\mathcal{X}=X)$, $H(Y|\mathcal{Z}=Z)$. We will remove $H(Y; X)$ and adopt well-defined notations to introduce task complexity, making the analysis easier to follow. Specifically, the classification task complexity for a specific dataset X is defined as: $H(\mathcal{T};X) = H(Y|\mathcal{X}=X)$. This clarification ensures that our theoretical analysis is built on clear notations and remains consistent with the existing work [1]. We will revise the notation and equations in the paper accordingly for improved clarity.
>
> **Concerns about assumptions**
>
> 1. How is it valid to assume Y follows a multivariate distribution?
>
> The assumption that the labels $\mathbf{Y}=\{Y_1,Y_2,Y_3,Y_4,… \}$  follow a multivariate distribution does not imply that we are predicting multiple labels simultaneously. Instead, we assume that the labels are drawn from a probabilistic distribution over the class labels, which reflects the uncertainty or the distributional properties of the labels in the task. In practice, the classifier is trained to predict a single label at a time, but the underlying distribution of the labels (such as a softmax distribution over categories) informs the decision-making process. The multivariate distribution helps capture the uncertainty of the labels in the model. This is consistent with the standard approach in classification tasks, where the output is a probability distribution over the classes, and the model is trained to maximize the likelihood of the correct label under that distribution.
>
> 2. We do not assume the independence of embeddings across samples.
>
> **Text notes**
>
> The distribution of $p$ is set to be uniform. In our implementation, we use torch.rand to generate it.
>
> Regarding the CNN layer, please combine Figure 2b with Table 8. Table 8 presents the CNN architecture used in the paper. The macro layer for the CNN is [0, 4], where each macro layer consists of multiple microlayers.
>
> **Concerns about positioning**
>
> The reviewer is correct in pointing out that noise injection techniques, such as dropout and diffusion models, have demonstrated benefits for model performance. We fully acknowledge these contributions and the positive impact of noise injection in these methods.
>
> In our work, we aim to extend the existing framework of positive noise [1] and introduce a new theoretical framework that explores the relationship between noise injection and task complexity in a more formal and quantifiable manner. We will revise the wording to more clearly reflect this context.
>
> [1] Poise-incentive noise, IEEE TNNLS, 2022.

---

### Review · Reviewer_5SGA · 2025-02-21

**Summary Of Contributions:**

This paper considers analyzing the impact of noise in machine learning from the perspective of task complexity measured by its entropy. The authors claimed that increasing task complexity can lead to better performances and considered different ways to inject noise to the learning process to increase task complexity. Experiments on different tasks, settings and architectures demonstrate the effectiveness of adding a specific type of positive noise (linear transformation noise) for improving model performances.



Requested changes:

**Audience:**

Yes

**Claims And Evidence:**

No

**Requested Changes:**

From my perspective, currently this submission is more like a technical report instead of a complete and inspiring research work. The scope of this work is ambiguous, and theoretical analysis and empirical results lack enough connections. Empirical results can also be substantially improved to better support and understand the proposed method. Detailed comments are listed below:

**Definition and scope of injected noise.** The authors simply considered three types of injected noise: Gaussian noise, linear transformation noise, and salt-and-pepper noise, without any justification on why these three types of noise are selected.

**Insufficient empirical results to support the broad claim.** I appreciate that the authors have conducted experiments with different image classification models (ResNet and ViT with different depths), different data sets and different settings (domain adaptation). However, this paper has a much broader claim for “learning systems”, which extends well beyond image classification tasks. It would be better if the authors can provide more experiments on other applications or restrict their claim to image classification tasks (possibly with text classification in Appendix).

**Theoretical analysis is very confusing.** The authors have provided many equations in their theoretical analysis, but it still makes me confused how such analysis can be useful for the proposed method. It seems that the authors simply provide some analysis on (under a much-simplified setup) how the task complexity may change if a specific type of noise is added to the training data. While the authors seem to provide a theoretically optimal design for the injected noise in Section 4.4, I wonder if such design is indeed applied to all experiments? The authors may need to clarify this part as well.

**More empirical results to fully understand the proposed method.** It seems that current experiments only show that adding specific linear transformation noise can be useful, and many other design factors are not fully investigated. For example, can the noise strength or layers added (as is compared in Figure 2) need to be adjusted during the training process? How will it affect the final performance compared to using the same setting? Some discussion and empirical comparison are certainly necessary here.

**Strengths And Weaknesses:**

Strength:
- The idea of adding “positive” noise during training is novel and interesting

Weakness:
- The authors failed to clearly explain the context of applicability of this research.
- Empirical results are not sufficient to fully understand and evaluate the proposed method

---

> ### Author Response · Authors · 2025-03-19
>
> Thank you for your valuable comments. We appreciate the time and effort you have taken to review our paper. Below is our response, and we will incorporate the required changes in the revision.
>
> 1. Gaussian noise is a widely used perturbation method that introduces random variations while preserving statistical properties. Linear transformation noise allows structured modifications. Salt-and-pepper noise, though commonly applied in image space, provides a non-continuous perturbation that helps analyze robustness against discrete disruptions. These choices offer a diverse set of noise injection strategies, covering both stochastic and structured perturbations. While these three types of noise have been widely used in existing works [1-4], our approach differs by injecting them into embeddings rather than raw images.
>
> 2. While the noise injection can be applied to broader learning systems, we recognize that validating it across diverse tasks would further strengthen our claim. To address this, we have included text classification results in Appendix F.9, demonstrating its effectiveness beyond image classification. Furthermore, existing works [5,6] have shown the effectiveness of noise injection in tasks such as semi-supervised learning and instruction fine-tuning for language models, reinforcing the broader relevance of our approach. In response to your feedback, we will refine our claims to align more closely with our experimental scope.
>
> 3. Our work extends the existing framework of positive noise [1] by introducing noise at the embedding level instead of the raw input space. The core concept in our analysis is task entropy, with Eqs. 4 and 5 being central to our formulation. Our theoretical analysis provides a guideline for incorporating different types of noise and offers a formal justification for how NoisyNN, when applied in a principled manner, can be beneficial. The theoretically optimal noise design in Section 4.4 is derived from the linear transformation case. We found a search space for linear transformation noise and proved the optimal design, given in Eq. 20, within this space. Additionally, we introduce an alternative circular shift design in Eq. 18. Empirical results for the optimal linear transformation noise are presented in Table 5, while results for the circular shift design are reported in Tables 1–4. We will clarify these distinctions in the revision.
>
>
> 4. Our theoretical analysis demonstrates that only linear transformation noise can reduce task entropy and improve performance. In Figure 2, which corresponds to the circular shift design (Eq. 18), our analysis predicts that the positive noise strength lies within the range of (0, 0.5). The experimental results in Figure 2 validate this prediction, showing consistency between theory and practice. A key advantage of our approach is that, rather than relying on extensive empirical searches for the positive noise strength, our theoretical framework provides a principled way to determine feasible noise ranges. Regarding layer selection, the results in Figure 2 indicate that injecting noise at deeper layers leads to better performance. Both our theoretical analysis and empirical results support the conclusion that applying noise at deeper layers and selecting the strength based on Eq. 18 is a good combination of design factors.
>
>  [1] Poise-incentive noise, IEEE TNNLS, 2022.
>
>  [2] Data Augmentation in Training CNNs: Injecting Noise to Images, arxiv, 2023.
>
>  [3] Variational Positive-incentive Noise: How Noise Benefits Models, arXiv, 2023.
>
>  [4] Enhance Vision-Language Alignment with Noise, AAAI 2025.
>
>  [5] NEFTune: Noisy Embeddings Improve Instruction Finetuning, ICLR 2024.
>
>  [6] InterLUDE: Interactions between Labeled and Unlabeled Data to Enhance Semi-Supervised Learning, ICML 2024.

---

### Decision · Action_Editor_3wSB · 2025-03-31

**Recommendation:** Reject

**Comment:**

The authors' rebuttal did not successfully address the reviewers' concerns, with two reviewers specifically noting that their questions remained unanswered. Ultimately, all three reviewers agreed that the submission does not meet the acceptance standards of TMLR.

**Audience:**

The topic is of interest in the field of robust and trustworthy deep learning. If sufficiently substantiated, the results may hold significance in improving our understanding of the role noise plays in vision classification tasks.

**Claims And Evidence:**

# Claims

This paper makes two major claims:

* a theoretical analysis of the impact of three types of noise on the image classification task, introducing the so-called positive noise, which is described as noise that reduces the complexity of the task.

* it proposes a method that achieves remarkable improvement on the ImageNet classification task by proactively injecting positive noise into the ViT and CNN architectures.

# Evidence
The first claim is supported by a subsection of the analysis, which builds on the previous work of (Li 2022), along with the math derivation presented in the Appendix.

The second claim is supported by empirical results, where a performance gain is achieved in top-1 accuracy on the ImageNet classification task, using the optimal quality matrix (Table 5).


# Reviewers' remark on the claims
All three reviewers agree that both claims are not sufficiently substantiated. Regarding the theoretical analysis (the first claim), two reviewers (`5SGA `and `yo8p`) identified a concerning gap between the theoretical analysis and the method. The reviewers also indicated that the analysis is very confusing, sloppy, or difficult to follow due to unclear notation.

As for the empirical results, while the reviewers acknowledge the improvement is significant, they demand additional experiments and more discussion to clarify where the performance gain originates.

The authors' rebuttal failed to persuade the reviewers. In particular, two reviewers noted that their questions were not addressed, leaving these major concerns unresolved.